# Uncovering Competency Gaps in Large Language Models and Their Benchmarks

## Abstract

The evaluation of large language models (LLMs) relies heavily on standardized benchmarks. These benchmarks provide useful aggregated metrics for a given capability, but those aggregated metrics can obscure (i) particular sub-areas where the LLMs are weak ("model gaps") and (ii) imbalanced coverage in the benchmarks themselves ("benchmark gaps"). We propose a new method that uses sparse autoencoders (SAEs) to automatically uncover both types of gaps. By extracting SAE concept activations and computing saliency-weighted performance scores across benchmark data, the method grounds evaluation in the model's internal representations and enables comparison across benchmarks. As examples demonstrating our approach, we applied the method to two popular open-source models and ten benchmarks. We found that these models consistently underperformed on concepts that stand in contrast to sycophantic behaviors (e.g., politely refusing a request or asserting boundaries) and concepts connected to safety discussions. These model gaps align with observations previously surfaced in the literature; our automated, unsupervised method was able to recover them without manual supervision. We also observed benchmark gaps: many of the evaluated benchmarks over-represented concepts related to obedience, authority, or instruction-following, while missing core concepts that should fall within their intended scope. In sum, our method offers a representation-grounded approach to evaluation, enabling concept-level decomposition of benchmark scores. Rather than replacing conventional aggregated metrics, CG complements them by providing a concept-level decomposition that can reveal why a model scored as it did and how benchmarks could evolve to better reflect their intended scope. Code is available at `anonymized`.

## 1 Introduction

Evaluating large language models (LLMs) relies heavily on benchmarks that report aggregated scores (e.g., accuracy or pass@k). Over the last decade, hundreds of benchmarks have been introduced across diverse domains [Guo et al., 2023; Chang et al., 2024]. While these benchmarks have fueled progress, uniform aggregation can obscure important sub-trends and mask model weaknesses [Hardt, 2025; Burnell et al., 2023]. For instance, Didolkar et al. [2024] disaggregated performance on MATH [Hendrycks et al., 2021a] and found topic-wise scores ranging from 27% to 74%, despite an overall score of 54%.

To counteract these aggregation issues, some benchmarks provide "semantic" topic annotations (e.g., hand-curated topics in MATH [Hendrycks et al., 2021b] or GPQA [Rein et al., 2024], or embedding-based clusters [Perez et al., 2023]). These high-level labels help characterize benchmark distributions and disaggregate performance, but they are coarse-grained and offer limited insight into model strengths and weaknesses. In particular, we lack a view of how finer-grained concepts, contexts, and reasoning patterns extend beyond coarse topic labels and how they relate to real-world model usage and capabilities [Miller and Tang, 2025; Mizrahi et al., 2024]. Furthermore, many of these semantic annotations are manually curated and difficult to scale. Without a scalable, fine-grained understanding of benchmark distributions, we risk overlooking benchmark gaps and systematically overtesting certain concept types.

In this work, we are interested in the automated identification of two types of gaps: (i) benchmark gaps, i.e., concept domains that are inadequately represented in an evaluation dataset, and (ii) model gaps, i.e., concept domains where models systematically underperform (see Figure 1). To this end,

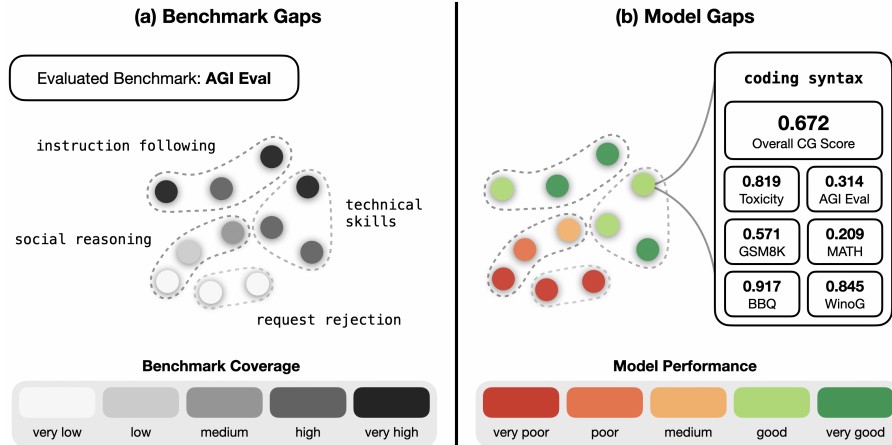

**Figure 1: Competency Gaps Overview.** Competency Gaps (CG) decomposes LLM evaluation into interpretable **benchmark gaps** and **model gaps** using the concept dictionary learned by a sparse autoencoder (SAE), a subset of which is visualized above. *(a) Benchmark Gaps* quantify how much each benchmark activates each concept, normalized by total dataset activation, and aggregated across benchmarks. *(b) Model Gaps* project model performance into concept space, assigning each concept per-benchmark and overall scores.

we introduce a new method called Competency Gaps (CG). Therein, we leverage sparse autoencoders (SAEs), which transform dense internal representations of a scrutinized LLM into high-dimensional, sparse feature vectors called SAE concept activations [Bricken et al., 2023; Cunningham et al., 2023]. Each dimension of these vectors is trained to capture a distinct, human-interpretable concept, assuming a sufficiently diverse and representative training distribution. We use pre-trained SAEs provided by the model authors, for which autointerpretability methods [McGrath et al., 2024] have already been applied to assign each dimension (or "concept") a textual label. This fixed set of concepts, often referred to as a concept dictionary, is defined by a pre-set dictionary size hyperparameter. This allows us to analyze the benchmark distribution over the SAE concept space [1] and identify benchmark gaps, and consequently map the model's performance onto the concept space to identify model gaps.

As a demonstration of the method, we evaluate Competency Gaps (CG) on two popular open-source models (`Gemma2-2B-Instruct` [Team et al., 2024] and `Llama3.1-8B-Instruct` [Grattafiori et al., 2024b]) across ten diverse benchmarks. We find notable gaps: benchmarks often miss concepts central to their intended scope while overtesting concepts tied to authority, control, and instruction-following. Additionally, we demonstrate how standard aggregated benchmark performance metrics tend to overly reflect the overrepresented concepts. Overall, this analysis serves to demonstrate the utility of CG as a method for uncovering and addressing gaps in prevailing evaluation paradigms. We summarize our contributions as follows:

- **Competency Gaps (CG) Method.** We introduce a systematic method for the automated identification of benchmark coverage and model performance gaps, using an SAE-based approach. The method can be applied to any LLM and benchmark of interest.[2] The method helps benchmark developers iteratively improve coverage, and enables model developers to gain fine-grained, distributional insights into model performance, as described in Figure 2.
- **Example Applications of CG on Popular LLMs and Benchmarks.** We applied CG to (`Gemma2-2B-Instruct` and `Llama3.1-8B-Instruct`) over ten diverse benchmarks, illustrating the kinds of insights the method enables: CG decomposed singular benchmark scores into interpretable axes derived from the models' own representations, surfaced consistent patterns of over- and under-tested concepts, and identified actionable improvements for both models and benchmarks.
- **Interactive Exploration Tool.** We release an open-source web application (Figure 6), alongside the code for the Competency Gaps method, for exploring per-concept model behavior and benchmark coverage. This enables users to audit model capabilities and benchmark balance in a transparent and interpretable way.

---

[1] Note that the proposed approach can only detect benchmark and model gaps for concepts that are represented in the SAE space. With more representative SAEs, concept coverage of our approach improves as well.

[2] While training an SAE specifically for the model at hand allows the analysis to be grounded in that model's own representations, we demonstrate that even LLMs without a dedicated SAE can be evaluated using CG by leveraging the SAE from another model. See Section 4.2.3.

**Workflow Recommendation:**
Evaluating a Model for a Target Capability
(using existing benchmarks)

1. **Select Initial Benchmarks.** Gather a broad set of candidate benchmarks that are potentially relevant to the capabilities you are interested in evaluating.

2. **Run Initial Analysis.** Execute the Competency Gaps analysis on the initial suite: calculate the cross-benchmark coverage score $\chi_{\text{bench}}^{(c)}$ and cross-benchmark model score $\chi_{\text{model}}^{(c)}$ for all concepts.

3. **Identify Gaps.** Use an LLM of choice to cluster the concepts identified as underrepresented or underperforming. Filter these clusters to find conceptual gaps that are relevant to your target profile.

4. **Iterate and Refine.** Augment your evaluation suite with new benchmarks that cover the identified gaps and retrain/fine-tune the model with data addressing the performance issues. Re-run the analysis and repeat the process until the coverage and performance aligns with your desired state.

**Workflow Recommendation:**
Building a Custom Benchmark

1. **Run Benchmark Analysis.** Implement a Hugging Face data loader for your benchmark and run the Competency Gaps analysis to identify all underrepresented and missing concepts.

2. **Synthesize Feedback.** Use an LLM of choice to filter concepts that are relevant to your use case and optionally ask it to cluster them into themes.

3. **Guide Data Generation.** Provide these themes as direct guidance for your data creation pipeline, whether it relies on human annotators or synthetic data generation.

4. **Augment and Iterate.** Add the newly created data to your benchmark and re-run the analysis. Repeat this iterative process until you are satisfied with the benchmark's conceptual coverage.

**Figure 2:** Recommended workflows for applying the Competency Gaps (CG) method in production.

## 2 RELATED WORK

**Weakness Identification in LLMs.** Identification of LLM weaknesses has evolved from anecdotal analysis to more systematic frameworks that break down performance into specific components [Jones and Steinhardt, 2022]. Among the first to do this were HarmBench [Mazeika et al., 2024] and garak [Derczynski et al., 2024], which established standardized evaluations of harmful behaviors. Various methods have introduced autoraters for this task: AutoDetect [Cheng et al., 2024], for example, used three LLM-powered agents to achieve a 50%+ weakness identification success rate. Gan et al. [2024] demonstrated that reasoning capabilities in LLMs can be systematically broken down and analyzed to identify specific weaknesses in logical inference chains. Other systematic evaluation methods have been proposed [Kim et al., 2023]; some work in this domain has also taken an adversarial learning approach [Yang et al., 2024].

**Cross-Model Behavior Comparison.** Beyond simple benchmark scoreboards, recent work has explored different approaches of characterizing the behavioral differences and internal representations across models and architectures in more detail [Zheng et al., 2025; Kim et al., 2025; Chang and Bergen, 2024]. Some works, including BehaviorBox [Tjuatja and Neubig, 2025]] and the LLM Comparator [Kahng et al., 2024], have emphasized side-by-side comparisons and actionable insights. Some mechanistic interpretability methods have emerged as well. One such method patches activations between model locations and decoding representations into interpretable text [Nada et al., 2024]. Recent work has also scrutinized the "universality hypothesis", i.e., universal features and circuits should appear for similar tasks across architectures [Shu et al., 2025; Yin et al., 2024].

**Benchmark Analysis and Down-Sampling.** Automated quality detection has advanced through frameworks like CLEAR [Chen and Mueller, 2024] and SMART Filtering [Gupta et al., 2024], automatically detecting and filtering problematic training data. A significant part of the work in this area has specifically focused on bias evaluation, a subset of quality assessment [Doan et al., 2024; Manerba et al., 2023; Koo et al., 2023]. Some preliminary work has also considered monitoring benchmark performance across time [Zhong and Raghunathan, 2025] or using meta learning [Calian et al., 2025].

**Comparison to Related Work.** In Appendix G, we compare our method to some of the above-mentioned methods that relate to the discovery of benchmark and model gaps. See Appendix G.1 for details of our comparison methodology, Appendices G.2 and G.3 for a high-level comparison of approaches and features of each method, and Appendices H, I, and J contain a comparison of results from each method.

# 3 COMPETENCY GAPS (CG) METHOD

In this section, we introduce **Competency Gaps (CG)**, an automated, SAE-based method that can be used to systematically evaluate and identify:

- **Benchmark gaps** given a set of evaluation benchmarks $\mathcal{B}$. How good is the coverage of various concepts in a specific or set of benchmarks? Do the benchmarks have any concept coverage gaps?

- **Model gaps** given a language model $M$ with an associated sparse autoencoder $SAE$. How well does the model perform across various concepts? What are its strengths and weaknesses?

Before diving into the specifics of the method, we first establish some notation. Each benchmark $b \in \mathcal{B}$ comes with an underlying dataset $D_b$. Furthermore, each concept $c \in C_{SAE}$ represents a distinct direction in the SAE space, and can be mapped to an autointerpretability label to which we refer as $l_c$. Based on SAE concepts, we introduce an SAE activation score $s_{c,i}$ that quantifies the degree of which concept $c$ was activated within a token sequence $x_i$.

> **SAE Concept Activation Score** $s_{c,i}$. A data point $i$ from a benchmark consists of an input token sequence $x_i$. The concept $c$'s activation on token $x_{i,j} \in x_i$ is computed via $SAE(x_{i,j}, c)$. We sum $SAE(x_{i,j}, c)$ over all tokens in $x_i$ to obtain the **SAE activation score** $s_{c,i}$ for concept $c$ on data point $i$. This is normalized by the length of the token sequence as $\tilde{s}_{c,i}$:
>
> $$s_{c,i} = \sum_{x_{i,j}} SAE(x_{i,j}, c), \quad \tilde{s}_{c,i} = \frac{s_{c,i}}{|x_i|} \tag{1}$$

Overall, our metrics were devised to satisfy: (i) All included benchmarks have an equal weight for computing the concept's cross-benchmark CG score, regardless of their size. (2) All data points have an equal weight for computing per-benchmark metrics, regardless of their token length.

## 3.1 BENCHMARK GAPS

Based on the introduced SAE concept activation score, we now introduce a measure to quantify concept coverage within and across benchmarks. These measures enable a distributional characterization of benchmarks, and consequently the identificaton of benchmark gaps.

> **Coverage Within an Individual Benchmark.** To evaluate the coverage of a concept $c$ in benchmark $b$, we define:
>
> $$\chi_{\text{bench}}^{(b,c)} = \frac{\sum_{i \in D_b} \tilde{s}_{c,i}}{\frac{1}{|C_{SAE}|} \sum_{c' \in C_{SAE}} \sum_{i \in D_b} \tilde{s}_{c',i}} \tag{2}$$
>
> which relates the ratio of the activation of SAE concept $c$ in dataset $b$ to the average concept activation in $b$.

> **Cross-Benchmark Coverage.** The overall coverage for concept $c$ in a benchmark suite $\mathcal{B}$ is the mean $\chi_{\text{bench}}^{(b,c)}$ across $\mathcal{B}_c$ (all benchmarks where $c$ is activated):
>
> $$\boldsymbol{X}_{\text{bench}}^{(c)} = \frac{1}{|B|} \sum_{b \in B_c} \chi_{\text{bench}}^{(b,c)} \tag{3}$$

**Coverage Classification.** We label a concept as *missing* from the benchmark suite if $\boldsymbol{X}_{\text{bench}}^{(c)} < \epsilon$, for some small $\epsilon$ (we used $e^{-5}$). Among the remaining concepts, those in the bottom decile (at or below the empirical 10th percentile) are *underrepresented*, and those in the top decile (at or above the empirical 90th percentile) are *overrepresented*. We can similarly define missing, underrepresented, and overrepresented concepts for each individual benchmark, using $\chi_{\text{bench}}^{(b,c)}$ instead.

## 3.2 MODEL GAPS

We now turn to the quantification of model gaps, and introduce a novel measure for model performance grounded in the discovered SAE concept space, both for individual and across benchmarks.

**Per-Benchmark Model Performance.** To evaluate model performance on a concept $c$ for a benchmark $b$, we define $\chi_{\text{model}}^{(b,c)}$:

$$\chi_{\text{model}}^{(b,c)} = \frac{\sum_{i \in D_b} m_b(i) \cdot \tilde{s}_{c,i}}{\sum_{i \in D_b} \tilde{s}_{c,i}} \tag{4}$$

where $m_b(i) \in [0,1]$ is the performance scoring policy for benchmark $b$ for datapoint $i \in D_b$ (normalized to [0,1]; higher is better). If no data points activate $c$ in $b$, then $\chi_{\text{model}}^{(b,c)}$ is undefined.

**Cross-Benchmark Model Performance.** The overall performance for concept $c$ in a benchmark suite $\mathcal{B}$ is the mean $\chi_{\text{model}}^{(b,c)}$ across all benchmarks:

$$\boldsymbol{X}_{\text{model}}^{(c)} = \frac{1}{|\mathcal{B}_c|} \sum_{b \in \mathcal{B}_c} \chi_{\text{model}}^{(b,c)} \tag{5}$$

We only consider benchmarks where $\chi_{\text{model}}^{(b,c)}$ is defined, i.e. $\mathcal{B}_c = \{b \in \mathcal{B} \mid \sum_{i \in D_b} s_{c,i} > 0\}$. If no data points in $\mathcal{B}$ activate concept $c$ (i.e., $|\mathcal{B}_c| = 0$), then $\boldsymbol{X}_{\text{model}}^{(c)}$ is undefined.

Given the cross-benchmark model performance score, we now introduce the notion of a model gap that pinpoints concepts associated with low model performance.

**Model Gap.** We label a concept a *model gap* if $\boldsymbol{X}_{\text{model}}^{(c)} < \epsilon$, for the same small $\epsilon$.

## 4 DEMONSTRATIONS OF THE METHOD

### 4.1 EXPERIMENTAL SETUP

**Benchmarks.** We demonstrate our method on ten static benchmark datasets, regularly used for performance and safety evaluations, and one arena-style benchmark. However, the presented method can be applied to any text-based benchmark.

- *Factuality:* Natural Questions [Kwiatkowski et al., 2019]; Vectara [Meyman, 2025].
- *Math:* GSM8K [Cobbe et al., 2021]; MATH [Hendrycks et al., 2021b].
- *Reasoning:* AGI Eval [Zhong et al., 2023]; LogicBench [Parmar et al., 2024]; SocialIQA [Sap et al., 2019]; WinoGrande [Sakaguchi et al., 2021].
- *Ethics & Bias:* BBQ [Parrish et al., 2021]; CrowS-Pairs [Nangia et al., 2020].
- *Arena Style:* LMSYS Chatbot Arena Zheng et al. [2023].

**Models.** We analyzed two popular open-source LLMs with available SAEs and autointerpretability labels. However, we would like to emphasize that the method is not bound to these particular models; it can be applied to any standard LLM with an SAE.

- `Llama3.1-8B-Instruct` + Goodfire SAE attached at layer 19 [Grattafiori et al., 2024a; Balsam et al., 2025].
- `Gemma2-2B-Instruct` + Gemma Scope SAE attached at layer 20 (residual stream) [GemmaTeam et al., 2024; Lieberum et al., 2024];[3]

---

[3] For the Goodfire Llama SAE, the choice of layer for SAE attachment was made by the creators of the SAE. We chose the Gemma Scope layer to be at a comparable depth. Generally, deeper layers have a tendency to represent higher-level, sentence- or discourse-level abstractions [Balcells et al., 2024; Shi et al., 2025].

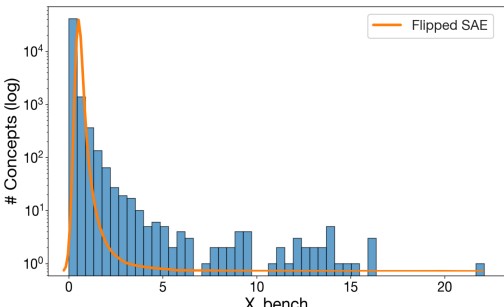 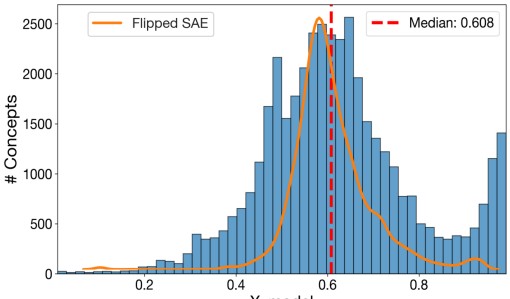

**Figure 3: Cross-Benchmark Coverage.** The distribution of $X^{(c)}_{\text{bench}}$ scores obtained for the 10 evaluated benchmarks, using the SAE of Llama 3.1 8B. This distribution exhibits strong left skew (most concepts have low coverage), and avg. performance is strongly dominated by a few concepts with high coverage (high $X^{(c)}_{\text{bench}}$). Similar skewed distributions were observed for individual benchmarks (Appendix Figure 12). The orange curve shows a similar analysis conducted through the activations and SAE concepts of Gemma 2 2B (see Section 4.2.3).

**Figure 4: Cross-Benchmark Model Performance.** The distribution of $X^{(c)}_{\text{model}}$ scores obtained for Llama 3.1 8B across the 10 evaluated benchmarks. The model exhibited high variance in performance across concepts. We observe particularly high performance for some concepts; these tend to include concepts related to coding, data handling, instruction following, and expressing positive sentiment toward the user. The orange curve shows a similar analysis conducted through the activations and SAE concepts of Gemma 2 2B (see Section 4.2.3).

## 4.2 RESULTS

To demonstrate the kinds of insights enabled by our method and exploratory web application, we report results of applying CG to `Llama3.1-8B-Instruct` across ten static benchmark datasets. This analysis can be applied to any language model and is intended as an active, iterative process in which both the benchmarking suite and the model are continuously refined (see Figure 2).

We report analogous results for `Gemma2-2B-Instruct` in Appendix B, and we further demonstrate how CG can be used with arena-style benchmarks by presenting results on the LMSYS Chatbot Arena in Appendix D. To efficiently sift through the large number of results that can arise from our method (e.g., given the scale of the SAE concept dictionaries), we sometimes use another LLM (Gemini 2.5 Flash, in this case) to filter, group, and summarize sets of concept descriptions.

### 4.2.1 BENCHMARK GAPS

**Existing Benchmarks Exhibit Skewed Representation Across Concepts.** The cross-benchmark coverage distribution across all ten benchmarks is shown in Figure 3. The distribution exhibits a strong left skew, resulting in both over- and under-representation of concepts in our (somewhat typical) suite of benchmarks. This skew results in any standard mean-based summary statistics being dominated by the outliers on the right part of the tail – a small number of "top concepts" with high representation (high $X^{(c)}_{\text{bench}}$). Conversely, when a concept has consistently *low* coverage across all included benchmarks, it risks being systematically under-tested. Another undesirable pattern is substantial overlap across benchmarks in a single evaluation suite, further illustrated in Figure 14 (Appendix C).

The top concepts mainly relate to starting new conversations and sports news, with a particular focus on football and sports achievements. Other prominent themes include syntax and attributes of articles. For example, the top 10 concepts by benchmark coverage included:

**(56130)** "English Premier League football discussions, especially about Manchester teams";

**(41290)** "New conversation or topic segment boundary marker".

Among the concepts with the *lowest* coverage score, one finds many concepts related to meta-cognition about the AI itself, e.g. its instructions, roleplaying boundaries, and how it discusses user inputs. The bottom 10 concepts by coverage include:

**(53553)** "The assistant should maintain professional boundaries when asked to roleplay"

**(25352)** "References to user messages or inputs in meta-discussion".

We identified 314 concepts (1%) as entirely missing from this particular suite of benchmarks (see Table 1). These again include concepts related to the AI's meta-cognition, as well as legal concepts.

**Individual Benchmarks Miss Relevant Concepts.** Individual benchmarks show a similar skew in their representation (Appendix Figure 12), and every benchmark except Vectara misses at least

| Concept ID | Concept Description |
|---|---|
| (2501) | The assistant explaining why it needs more information |
| (2641) | The assistant needs to explain its limitations or capabilities |
| (2009) | Regulatory classification and compliance requirements |

**Table 1: Examples of Missing Concepts from the Full Benchmark Suite.**

30% of all concepts (Appendix Figure 13). However, benchmarks are usually designed to evaluate a specific subset of capabilities, and so of course it may not be desirable for individual benchmarks to have complete coverage of all concepts.

More importantly, we would like benchmarks to have coverage of *relevant* concepts. To identify such missing relevant concepts, we first used the CG method to identify all missing concepts for a given benchmark. Then, we used an LLM (Gemini 2.5 Flash, in this case) to find those concepts that one might expect to be in scope for the benchmark (see Appendix **??** for the prompt). We also used the open-sourced web app to explore and verify these examples. This process exemplifies our recommendation to use the CG method for unsupervised discovery of coverage gaps.

Table 2 presents illustrative examples of concepts that are missed by benchmarks. These are concepts that seem central to the desired goals of the benchmarks, e.g. "The need for thorough and objective assessment of evidence" was missing from *AGIEval*, and "Instructions about how someone should behave or what qualities to embody" was missing from *SocialIQA*.

| Benchmark | Concept ID | Concept Description |
|---|---|---|
| *AGIEval* | (33456) | The need for thorough and objective assessment of evidence |
| | (59559) | Careful qualification and nuanced explanation of complex topics |
| *LogicBench* | (56997) | The model is explaining how different elements or factors relate to each other |
| | (11957) | Mathematical and logical concepts across multiple languages |
| *SocialIQA* | (35877) | Speaker defending or explaining their planned actions against expectations |
| | (1897) | Instructions about how someone should behave or what qualities to embody |

**Table 2: Examples of Missing Concepts from Three Individual Benchmarks.**

### 4.2.2 MODEL GAPS

In addition to benchmark gaps, we also analyze model performance gaps – on which SAE concepts do the models perform particularly well or particularly poor?

**Model's Best-Performing Concepts Include Commitment to Help, Coding.** Because the SAE concepts are fixed for a given model, our method enables us to compare and create composite results across benchmarks. Figure 4 shows the distribution across concepts for cross-benchmark model performance, which shows that the model performs well on a number of concepts, with high performance across all benchmarks. Concepts with the highest cross-benchmark performance tended to concern engineering related tasks (i.e., coding or data handling) or helpful behaviors (positive sentiments towards the user, or delivering an accurate response). The top 10 concepts included:

(20022) "Iteration or traversal through sequences in programming",

(24074) "The assistant is about to provide an illustrative example",

(2461) "Assistant expressing commitment to help or do its best".

**Model's Worst-Performing Concepts Include Polite Rejection, Time, and Setting Boundaries.** Perhaps more critical, from the standpoint of model evaluation, are the concepts which attained the poorest overall performance across benchmarks. An interesting recurring theme is that the worst-performing concepts include opposites to the helpful/sycophantic concepts discussed in the previous section (which came at the top in performance). Examples of worst-performing concepts include:

(26535) "The assistant needs to politely reject or redirect inappropriate requests",

(56928) "The assistant maintaining professional boundaries while offering appropriate help".

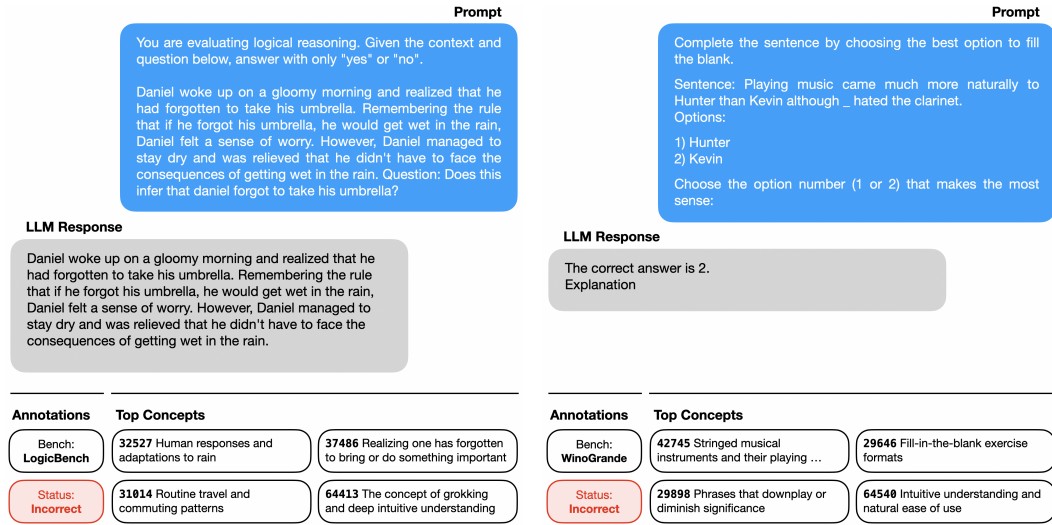

**Figure 5: Model Gap Illustrated on Specific Benchmark Datapoints.** Example LogicBench and WinoGrande items associated with "intuitive understanding" concepts (left: 64113, right: 64540). Llama 3.1 8B answered both incorrectly, consistent with these concepts being model gaps.

Furthermore, Competency Gaps identified other groups of competencies that have been anecdotally identified as LLM weaknesses in prior literature, validating our automated approach as a scalable and systematic method for identifying such model weaknesses. These include:

- **Representations of Time:**
    **(29324)** "Historical date and time period formatting"
    **(12644)** "Cooking time durations in recipe instructions")
- **Image Manipulations:**
    **(30206)** "Image contrast adjustments in photo editing and computer vision" [4]
- **Palindromes / Reasoning over Letters:**
    **(56613)** "Code examples and explanations of palindrome checking algorithms"
- **Mathematical Operations:**
    **(64527)** "Mathematical addition operator in calculations".

Additionally, the proposed method surfaces LLM weaknesses that have *not* been previously studied in the literature. One such category is "appeals to intuition in reasoning or decision making ":

**(64413)** "The concept of grokking and deep intuitive understanding"

**(64540)** "Intuitive understanding and natural ease of use"

Moreover, the exploratory web application allows users to directly examine example data points and better understand how such competencies manifest in practice (see Figure 5).

### 4.2.3 ROBUSTNESS

Prior work has provided mixed evidence on the stability of SAE concepts [**??**]. We therefore set out to evaluate the stability and generalizability of CG findings.

**A Model-Specific SAE Is Not Necessary, and Different SAEs Can Yield Similar Results.** We tested whether the CG insights derived from Llama 3.1 8B using its own model-specific SAE align with those obtained through Gemma 2 2B activations and SAE. As shown in Figures 3 and 4, the overall shape and medians of the score distributions were similar, especially considering the difference in dictionary sizes (Llama's SAE contains 2.8x more concepts than Gemma's). When comparing the best- and worst-performing concepts, we observed clear correspondences and similar interpretive

---

[4]`Llama3.1-8B-Instruct` is a text-only model, and we found that such concepts were activated by metadata indicating image editing, Photoshop scripts, and tutorials on the topic. This type of review of specific data points—both in the benchmarks and in the SAE training dataset—is enabled by our web app, and proves helpful for understanding the context and nuance of various concepts.

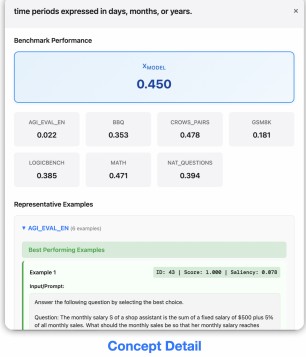

**Figure 6: Exploratory Web Application Overview.** The interface presents a searchable and filterable list of all concepts in the *Main Concept Overview*, with an expandable *Concept Detail Modal* that provides additional per-benchmark information, including specific example generations. The application also includes dedicated sections for cross-benchmark correlation, benchmark coverage inspection, and supplementary analyses. Additional screenshots are provided in Appendix F.

insights, as shown in Table 3. This suggests that CG can yield meaningful insights even for LLMs *without* their own pre-trained SAE and demonstrates the overall stability of the method. Nonetheless, we expect that a model-specific SAE with a larger dictionary still offers the most precise and grounded results.

| Analysis | Llama SAE | Gemma SAE |
|---|---|---|
| *Best Performance* | **(45314)** Legal reasoning and argumentation patterns in multiple choice questions 
 **(27510)** Code patterns for including JavaScript resources in web pages | **(5471)** References to legal cases and procedural aspects of law 
 **(8196)** Code constructs or reserved keywords in programming languages |
| *Worst Performance* | **(2874)** Mathematical differentiation operators and notation 
 **(2872)** Explaining time requirements and duration | **(13908)** Numerical values, counts or measurements 
 **(9936)** Dates and numeric sequences |
| *Best Coverage* | **(41290)** New conversation or topic segment boundary marker 
 **(902)** Step-by-step mathematical explanations and calculations | **(11527)** The start of a document 
 **(11880)** Mathematical expressions and calculations related to derivatives and factors |
| *Worst Coverage* | **(27900)** Discussions of factual accuracy and consistency checking 
 **(47946)** The assistant explains how it processes and handles information | **(5657)** Terms related to correctness and accuracy in responses or answers 
 **(1797)** Phrases related to instructions or operational processes |

**Table 3: Examples of Concept Correspondences in Best/Worst Performance and Coverage Analyses.** The analysis was conducted for Llama 3.1 8B: the Llama SAE results used the model's own activation and custom SAE; the Gemma SAE results used Gemma 2 2B's activations and SAE.

**CG Scores Are Consistent Across Perturbations** To assess the robustness of CG scores, we re-ran the full analysis 100 times, each time randomly dropping 20% of the examples per benchmark. The resulting standard deviations were low: 0.014 for $X_{\text{model}}$ and 0.025 for $X_{\text{bench}}$ on Llama. This indicates that CG yields stable scores under random subsampling.

**CG Scores Respond to Adversarial Perturbations.** We conducted an adversarial ablation in which we identified the top 100 best- and worst-performing concepts, then removed the most salient 100 datapoints associated with them across all benchmarks. As expected, removing rows aligned with high-performing concepts lowered median $X_{\text{model}}$ on average by 0.6%, while removing those aligned with low-performing concepts increased it on average by 1.3% (repeated across 10 repetitions). Despite removing less than 1% of the testing data, we were able to make predictable and consistent changes to the overall performance, suggesting that CG surfaces meaningful concept-level information.

## 5 DISCUSSION

Our analysis revealed a structural imbalance in a sample of popular benchmarks. For example, the benchmarks strongly emphasized concepts related to authority, control, and instruction-following, while neglecting complementary concepts related to polite refusals, meta-cognition, and meta-discussion about the AI itself. When evaluating across a benchmark suite, such skewed representation within the benchmarks may skew our perception of model capabilities. Our method also identified potential coverage gaps in specific benchmarks, pinpointing missing concepts that seemed relevant to each benchmark's scope.

A similar bias toward sycophancy and instruction-following emerged in the model gap analysis. Here, positive or sycophantic concepts score highest, while opposing concepts (such as those linked to rejecting requests or setting boundaries) score lowest. While this is likely due in part to instruction-based post-training, we note that model gaps and benchmark gaps are heavily intertwined, and that benchmark gaps may lead to model gaps — model developers may unknowingly overlook (or be disincentivized to address) model weaknesses that are poorly covered by existing benchmarks.

### 5.1 LIMITATIONS

**Concept Coverage is Limited to SAE Concept Space.** The usage of SAE concepts implies that we can only detect competency gaps for concepts that have *some* representation in the model, as captured by the SAE dictionary. Thus, we should take care in how we interpret our results: (1) *benchmark gap* analyses reveal where models possess internal representations for concepts that evaluations fail to adequately test, and (2) *model gap* analyses identify cases where models have flawed or partial internal representations that they cannot apply effectively to downstream tasks. However, it should be noted that with more representative SAEs, the concept coverage of our proposed approach improves as well.

**Concept Labels.** In addition to being limited to existing SAE concepts, as mentioned above, our method inherits some limitations of SAEs. These include a lack of ground truth [Smith et al. [2025]] (relatedly, the autointerpretability labels are automatically generated, though our web app enables easy spot checks as needed), non-convergence when SAEs are retrained [Paulo and Belrose, 2025], "feature absorption" (where token-aligned latents "absorb" some expected feature directions, and the tendency for SAEs toward learning common feature combinations instead of atomic features) [Chanin et al. [2024]; O'Neill et al. [2025]]. Despite these limitations, SAE remain a well regarded mechanistic interpretability method for unsupervised hypothesis discovery, which is how we use them in the current work.

### 5.2 DOWNSTREAM APPLICATIONS AND FUTURE WORK

**Benchmark Search and Selection.** Our method could be integrated into a database of available benchmarks (e.g., Hugging Face Datasets). Users seeking benchmarks to evaluate their models could use CG to inform their selection to achieve a desired coverage. Similarly, creators of benchmarks or the database maintainers could use CG to inform future benchmark creation.

**Targeted Creation of Novel Benchmark Data.** Beyond characterizing existing benchmarks, a list of underrepresented concepts within the suite could guide the generation of novel benchmark data. Such data could be created either by prompting LLMS with autointerpretability labels or by directly applying SAE-based steering interventions during generation.

**Method Improvements.** We hope that this work may serve as a starting point for further development on the method. For example, in future we may wish to incorporate an automated sensitivity analysis for the choice of SAE layer, or per-token activations for more fine-grained analyses.

Current model evaluations may risk a narrower view of "competence," potentially leaving critical gaps untested. We hope that our method enables benchmark developers and model evaluators to identify both benchmark gaps and model gaps, uncovering and addressing model weaknesses in areas that may be essential for real-world, human-facing use-cases.

## REPRODUCIBILITY STATEMENT

To maximize reproducibility of our work, both the analysis and visualization code is open-sourced at anonymized. This repository also includes the data extracted for the models and datasets evaluated in this paper. Furthermore, all prompts used for LLM clustering are included in Appendix E and the setup of alternate methods is described in Appendix G.1.

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

# A LLM USAGE

LLMs were used in parts of the implementation and during the writing of the paper (e.g., paragraph shortening, transition refinement, etc.). AI-powered search engines were also used to help identify some references. LLM clustering was used to sift through large amounts of data produced by our method.

# B ADDITIONAL RESULTS: GEMMA 2 2B INSTRUCT

## B.1 BENCHMARK GAPS

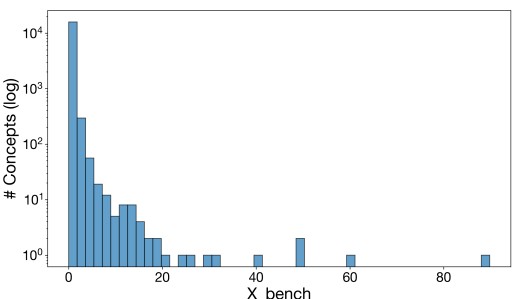

**Figure 7: Cross-Benchmark Coverage.** The distribution of $X_{\text{bench}}^{(c)}$ scores obtained for the 10 evaluated benchmarks, using the SAE of Gemma 2 2B.

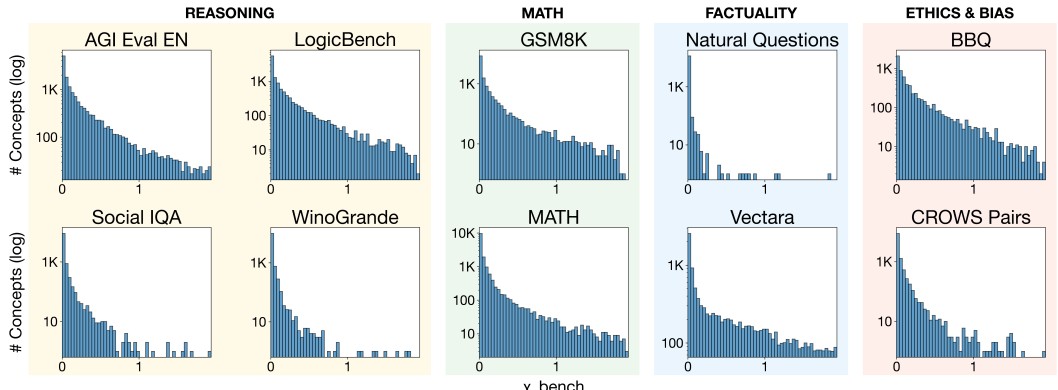

**Figure 8: Coverage Within Individual Benchmarks.** A breakdown of $\chi_{\text{bench}}^{(b,c)}$ score distributions for individual benchmarks obtained via Gemma 2 2B.

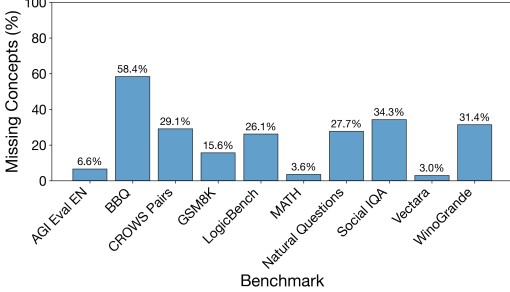

**Figure 9: Missing Concepts.** Proportion of the SAE concept dictionary for Gemma 2 2B that is not tested by the respective benchmarks.

| Concept ID | Concept Description |
|---|---|
| (5169) | patterns of repeated characters or symbols |
| (3674) | phrases related to coding or programming syntax |
| (9514) | detailed references and citations in academic writing |
| (10108) | questions and references to uncertainty or confusion |
| (5102) | specific programming or technical terminology related to data storage and handling |

Table 4: Examples of Missing Concepts from the Full Benchmark Suite.

| Benchmark | Concept ID | Concept Description |
|---|---|---|
| *AGIEval* | (12792) | phrases related to problem-solving or troubleshooting |
|  | (12436) | error handling and debugging statements in programming code |
| *LogicBench* | (7873) | phrases that indicate conditions or states related to certainty or necessity |
|  | (7264) | occurrences of mathematical or formal logic terms and control structures in the text |
| *SocialIQA* | (2863) | concepts related to social dynamics and collaborative efforts |
|  | (12897) | concepts related to socio-cultural analysis and individualized experiences |

Table 5: Examples of Missing Concepts from Individual Benchmarks.

## B.2 MODEL GAPS

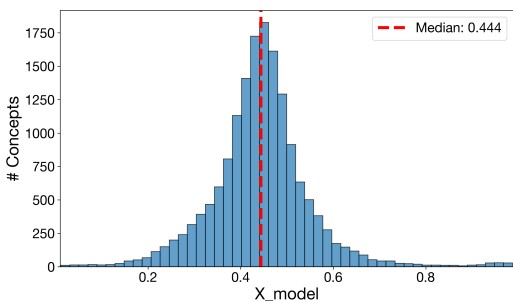

**Figure 10: Cross-Benchmark Performance.** The distribution of $X^{(c)}_{\text{model}}$ scores obtained for for Gemma 2 2B.

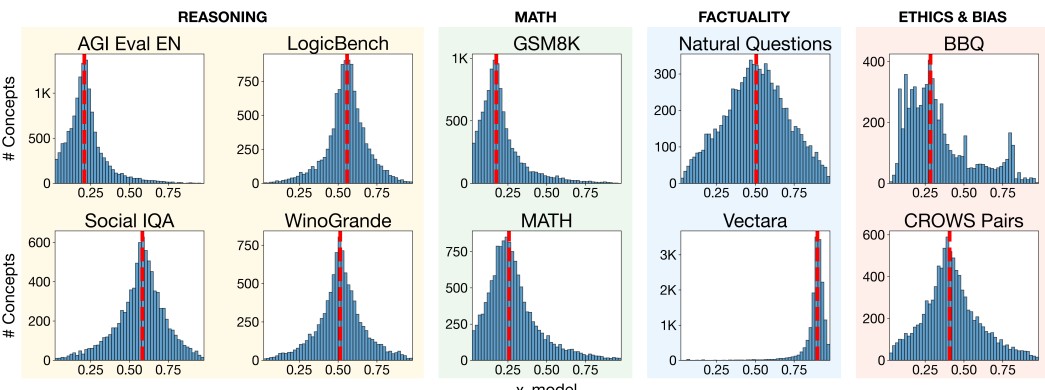

**Figure 11: Per-Benchmark Distributions for Model Performance.** A breakdown of model performance $\chi_{\text{model}}^{(b,c)}$ score distributions for individual benchmarks obtained for Gemma 2 2B. The red line indicates the median.

### B.3 ROBUSTNESS

**Perturbation Consistency.** We re-ran the full analysis 100 times, each time randomly dropping 20% of the examples per benchmark. The resulting standard deviations were: 0.012 for $X_{\text{model}}$ and 0.011 for $X_{\text{bench}}$.

**Adversarial Perturbations.** Upon removal of the most salient 100 datapoints associated with top 100 best-performing concepts across all benchmarks lowered median $X_{\text{model}}$ on average by 0.8%. On the other hand, removing the most salient 100 datapoints associated with top 100 worst-performing concepts across all benchmarks increased median $X_{\text{model}}$ on average by 0.5%. This process was repeated 10 times.

# C   ADDITIONAL RESULTS: LLAMA 3.1 8B INSTRUCT

## C.1   BENCHMARK GAPS

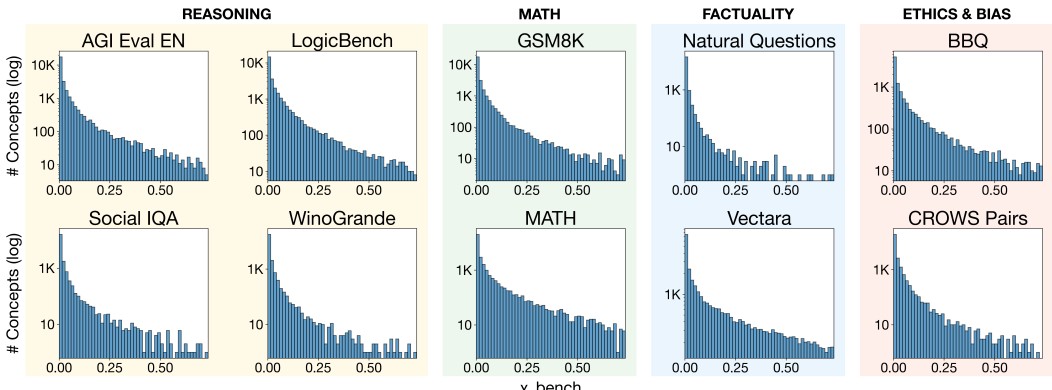

**Figure 12: Coverage Within Individual Benchmarks.** A breakdown of $\chi_{\text{bench}}^{(b,c)}$ score distributions for individual benchmarks obtained via Llama 3.1 8B. These distributions all show strong left skew, such that average performance on each benchmark is strongly dominated by a small number of concepts with high coverage (high $\chi_{\text{bench}}^{(b,c)}$).

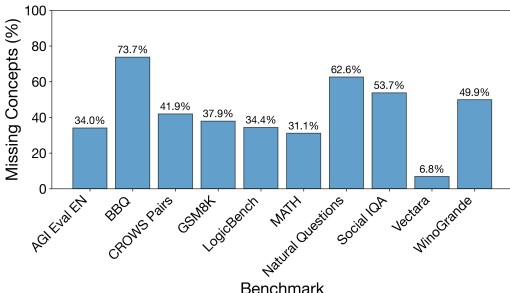

**Figure 13: Proportion Missing Concepts, for Individual Benchmarks.** Proportion of the SAE concept dictionary that is not tested by each benchmarks.

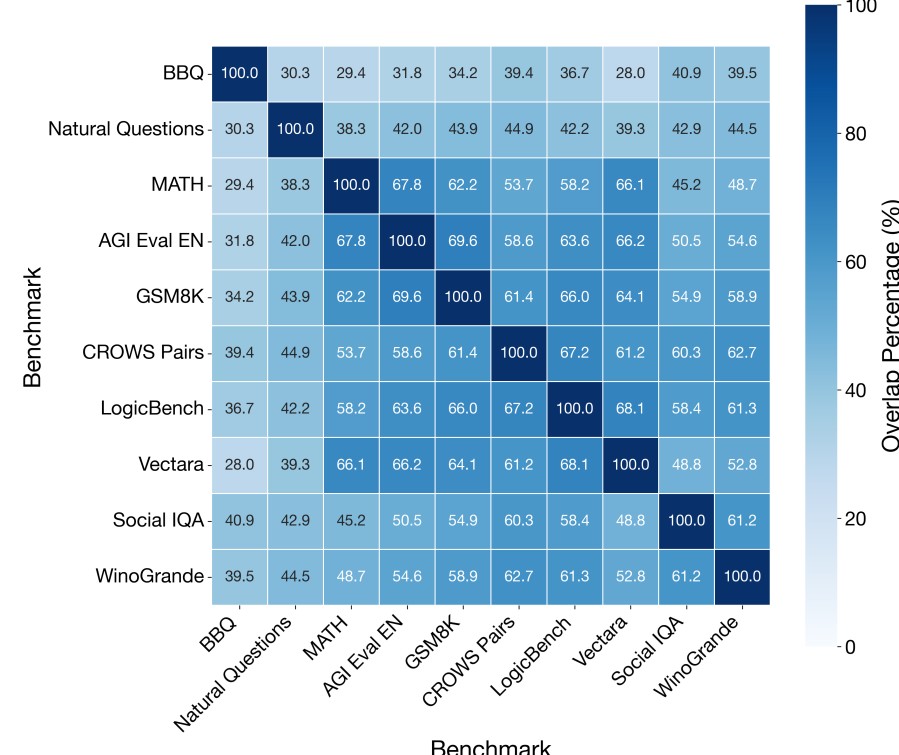

**Figure 14: Cross-Benchmark Concept Overlap.** Jaccard similarity of $X_{\text{bench}}^{(c)}$ coverage profiles between benchmark pairs, obtained through Llama 3.1 8B, showing which benchmarks share similar concept coverage.

## C.2  MODEL GAPS

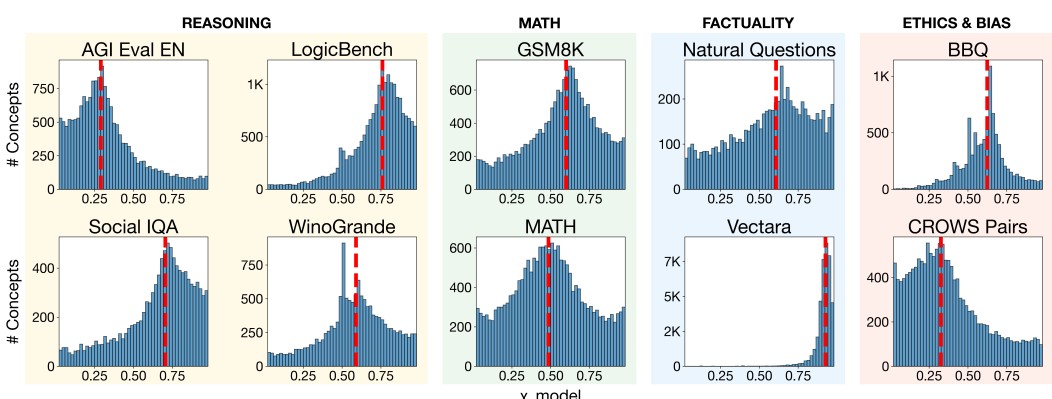

**Figure 15: Per-Benchmark Distributions for Model Performance.** A breakdown of model performance $\chi_{\text{model}}^{(b,c)}$ score distributions for individual benchmarks obtained for Llama 3.1 8B. The red line indicates the median.

# D  ADDITIONAL RESULTS: LMSYS CHATBOT ARENA

To illustrate how our method can be applied to arena-style benchmarks, we apply CG on Llama 3.1 8B with LMSYS Chatbot Arena Zheng et al. [2023].

## D.1  IMPLEMENTATION DETAILS

Unlike the rest of the benchmark datasets evaluated in this paper, which are static datapoints with a scoring policy, LMSYS Chatbot Arena Zheng et al. [2023] relies on preference annotations from humans presented with responses of two LLMs at a time, competing on the same input. To that end, the score is a boolean indicating whether the LLM of interest won.

We use the data from https://huggingface.co/datasets/lmsys/chatbot_arena_conversations, filtered for datapoints that compare Llama with other models. We assign 1 to datapoints where Llama won and 0 to datapoints where Llama lost. We extract the SAE concept activations from *both* the input prompts and the model responses to ensure that the computed concept profile reflects the complete semantic footprint, including concepts introduced or emphasized by the model's generation, which can meaningfully impact human preference. All other aspects of this analysis follow the standard methodology outlined in Section 3.

## D.2  BENCHMARK GAPS

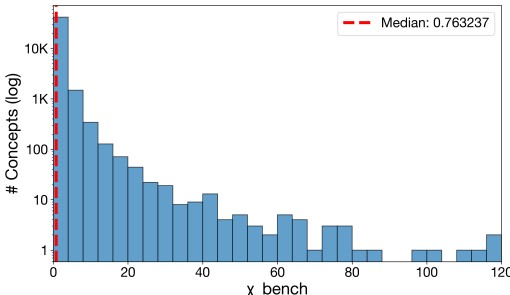

**Figure 16: Benchmark Coverage.** The distribution of $\chi_{\text{bench}}^{(b,c)}$ scores obtained for LMSYS Chatbot Arena, using the SAE of Llama 3.1 8B.

| Benchmark | Concept ID | Concept Description |
|---|---|---|
| *Best Coverage* | (902) | Step-by-step mathematical explanations and calculations |
| | (9287) | Numbered steps in instruction lists and process descriptions |
| *Worst Coverage* | (27900) | Discussions of factual accuracy and consistency checking |
| | (14146) | The assistant should reject inappropriate or NSFW requests |

**Table 6: Examples of Specific Concepts with the Best and Worst Coverage in LMSYS Chatbot Arena, Obtained Through Llama 3.1 8B.**

### D.3 MODEL GAPS

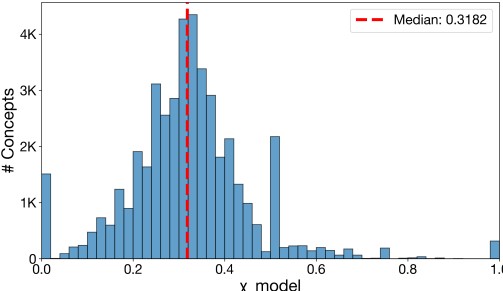

**Figure 17: Benchmark Performance.** The distribution of $\chi_{\text{model}}^{(b,c)}$ scores obtained for LMSYS Chatbot Arena, using the SAE of Llama 3.1 8B.

| Benchmark | Concept ID | Concept Description |
|---|---|---|
| *Best Performance* | (2691) | Multiple choice format with options A (okay), B (good), C (wrong) for evaluating behaviors |
| | (45314) | Legal reasoning and argumentation patterns in multiple choice questions |
| *Worst Performance* | (27171) | The assistant is breaking down complex topics into fundamental concepts |
| | (52258) | Password-related security discussions and requests |

**Table 7: Examples of Specific Concepts within Llama 3.1 8B with the Best and Worst Performance on LMSYS Chatbot Arena.**

## E    PROMPTS FOR LLM ANALYSIS OF CG RESULTS

To help filter through a large number of results, we used an LLM (Gemini 2.5 Pro) to sift through results (see Appendix A). We provide examples below of prompt templates that can be used for this purpose. Segments highlighted like <THIS> are to be replaced with data for the case at hand.

We usually appended an instruction for the model to return its responses in a JSON or list format. Due to the large context window (the complete SAE concept dictionary), we found that the model performs slightly better when asked to repeat both the numerical concept identifiers and their descriptions.

### E.1    BENCHMARK GAPS: MISSING CONCEPTS (CROSS-BENCHMARK)

Below is a list of concepts in a large language model. Each concept has an ID and a description. Are any of these concepts *critical* to the evaluation of large language models? Such concepts generally span topics of safety (toxic language, harm, bias, etc.), performance (reasoning ability, math, coding, etc.), and metacognition (ability to reject responses, reasoning about instructions, etc.). Choose from the list of concepts below. List all such relevant concepts. Do not summarize or group; list all concepts verbatim as they appear below if they are relevant.

LLM CONCEPTS:
<AVAILABLE_CONCEPTS>

### E.2    BENCHMARK GAPS: MISSING CONCEPTS (PER-BENCHMARK)

Below is a list of concepts in a large language model. Each concept has an ID and a description. Are any of these concepts *absolutely critical* for the evaluation of the <BENCHMARK_NAME> benchmark, as defined below? Choose from the list of concepts below. List all such relevant concepts. Do not summarize or group; list all concepts verbatim as they appear below if they are relevant.

BENCHMARK DEFINITION:
<BENCHMARK_DEFINITION>

LLM CONCEPTS:
<AVAILABLE_CONCEPTS>

### E.3    BENCHMARK GAPS: MATCHING

Below, there is (1) a list of Competency Gaps concepts and (2) a list of <OTHER_FRAMEWORK> categories.

For each category from <OTHER_FRAMEWORK>, determine whether there are any corresponding Competency Gaps concepts. If no relevant concepts exist, leave this blank.

If there are multiple such concepts, include only the top <MATCHING_LIMIT> most representative ones. Do not include more than <MATCHING_LIMIT> concepts per category.

(1) COMPETENCY GAPS CONCEPTS:
<AVAILABLE_CONCEPTS>

(2) <OTHER_FRAMEWORK> CONCEPTS:
<OTHER_FRAMEWORK_CONCEPTS>

## F EXPLORATORY WEB APPLICATION: ADDITIONAL SCREENSHOTS

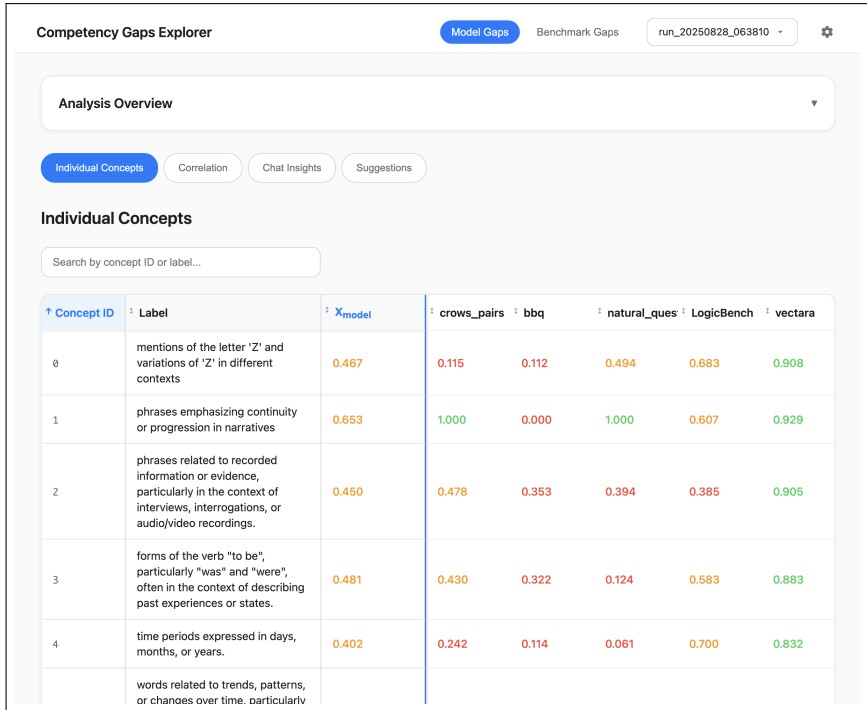

**Figure 18:** Web Application Screenshot: An overview of all concepts for the Model Gaps analysis.

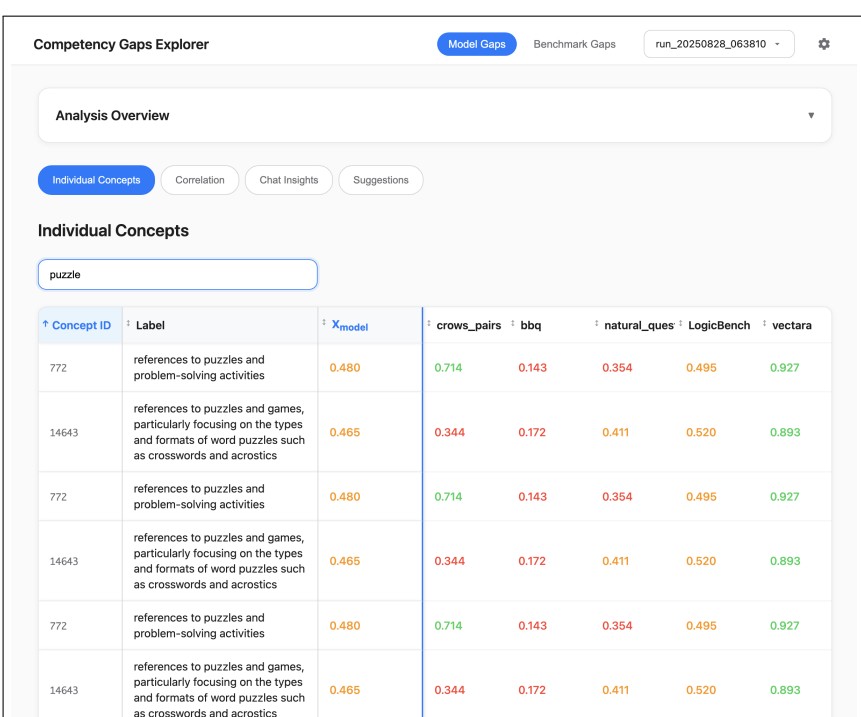

**Figure 19:** Web Application Screenshot: Keyword-filtered concepts for the Model Gaps analysis.

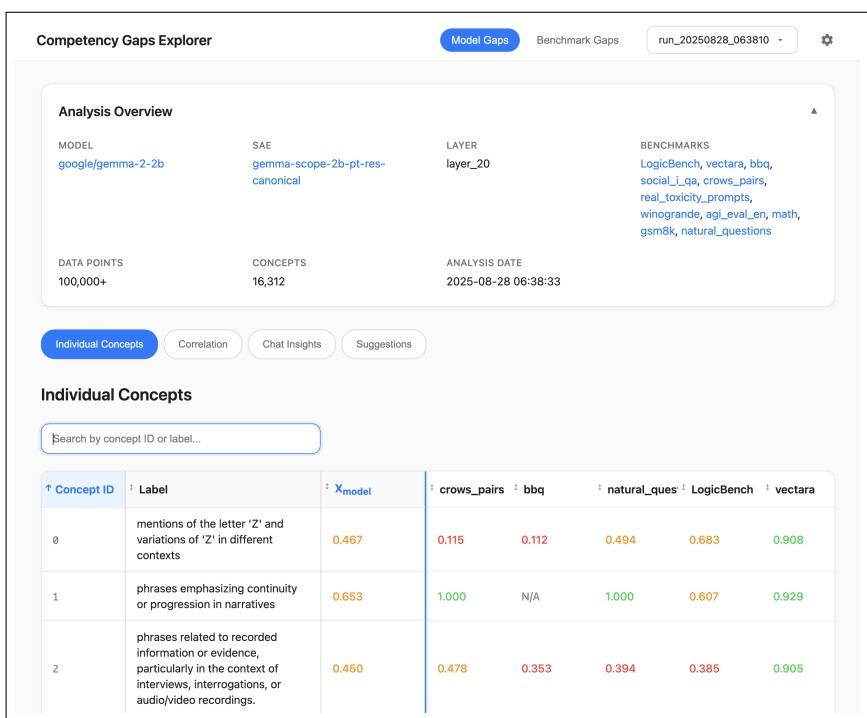

**Figure 20:** Web Application Screenshot: Expandable view with the analysis metadata.

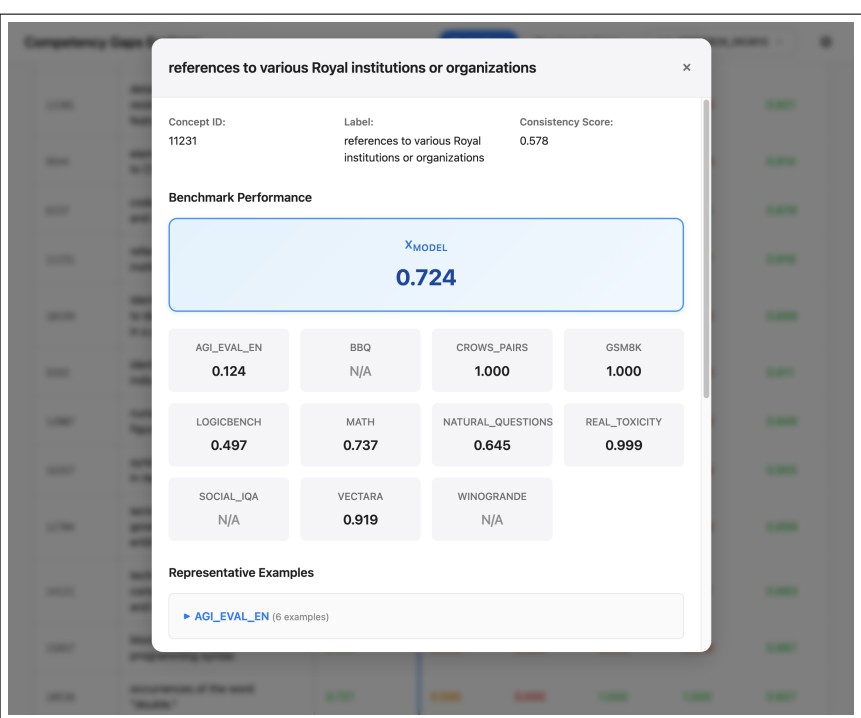

**Figure 21:** Web Application Screenshot: Concept detail within the Model Gaps analysis, summarizing the performance of this concept across benchmarks.

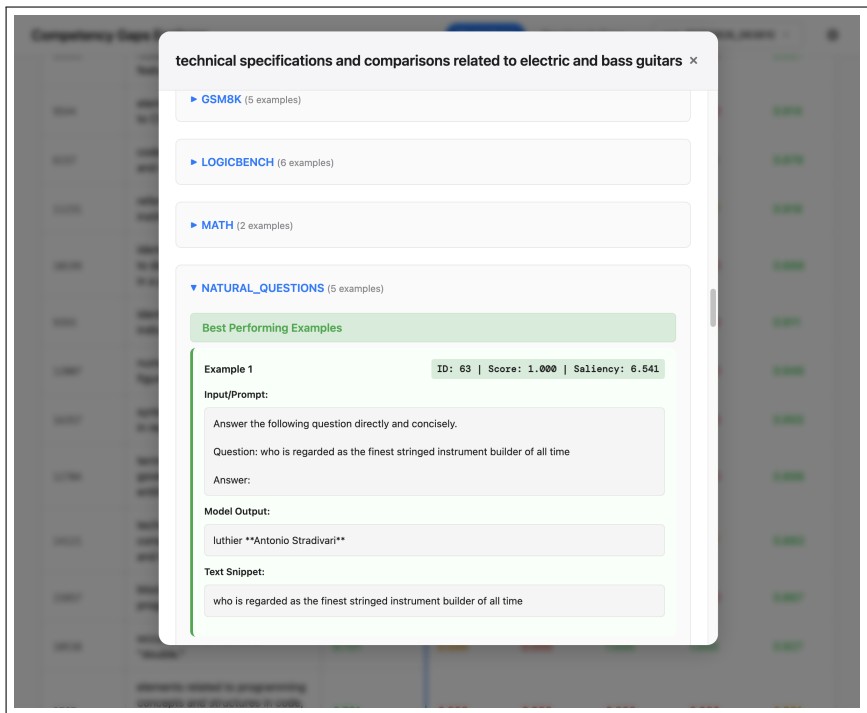

**Figure 22:** Web Application Screenshot: Examples of data points where the model performed well and the concept at hand shows high activation.

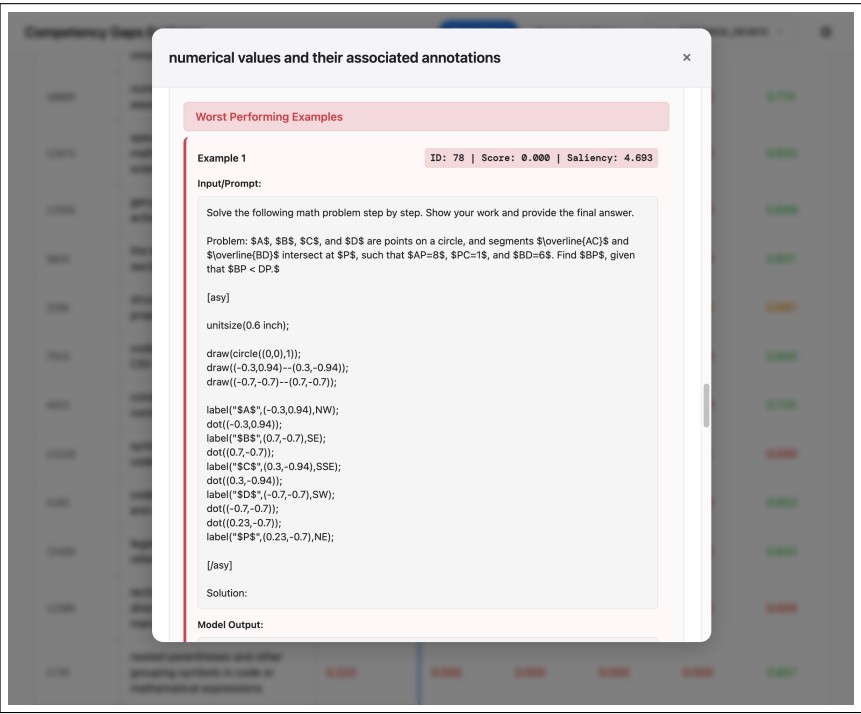

**Figure 23:** Web Application Screenshot: Examples of data points where the model performed poorly despite the concept at hand showing high activation.

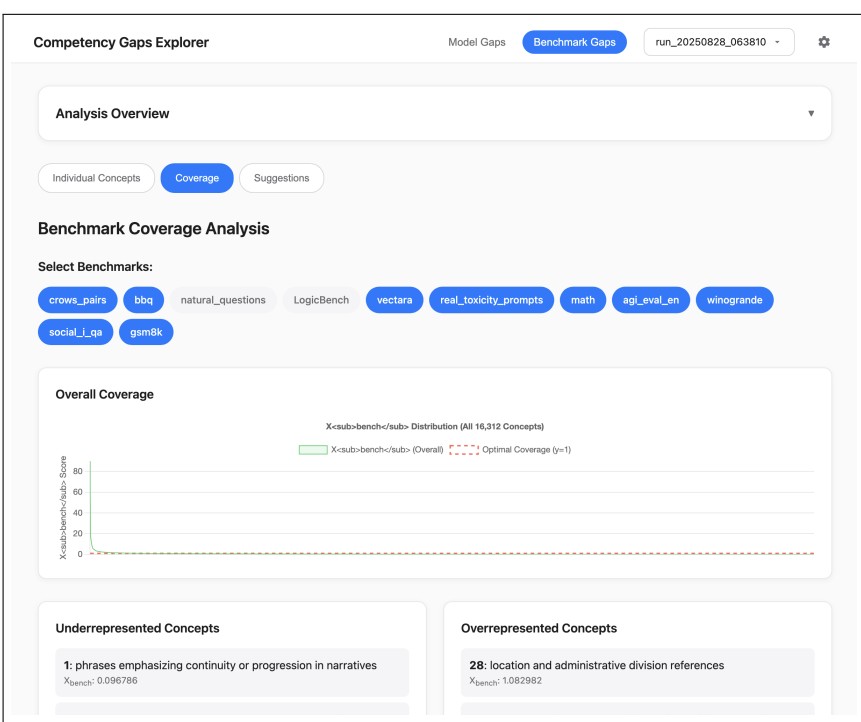

**Figure 24:** Web Application Screenshot: Coverage visualization comparing the coverage and distribution of concepts across different combinations of analyzed benchmarks.

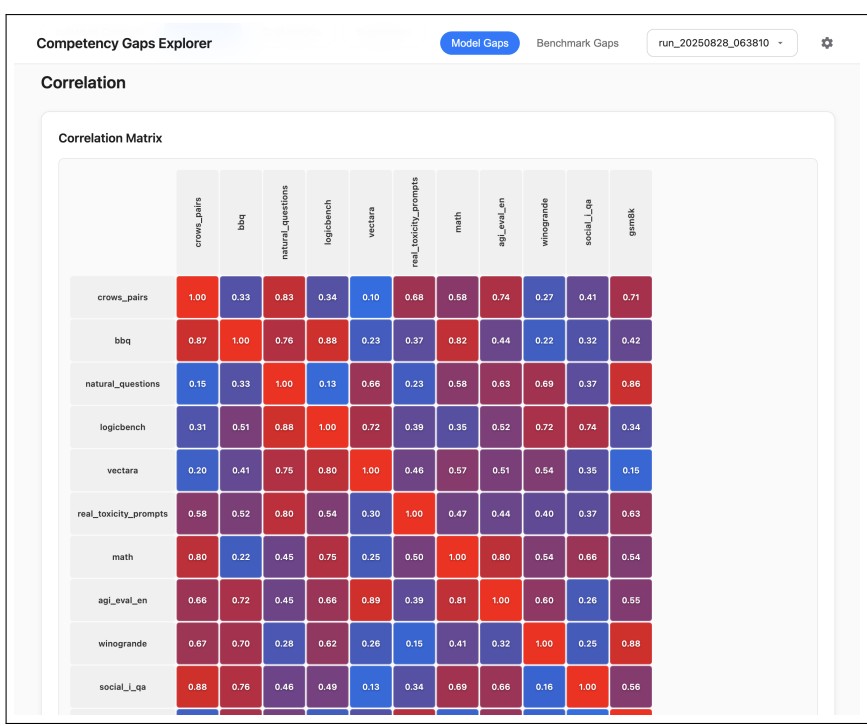

**Figure 25:** Web Application Screenshot: Grid showing the correlation of scores across benchmarks.

# G   COMPARISON WITH OTHER METHODS

## G.1   COMPARISON METHODOLOGY

We compare our method to some of the related methods from Section 2 – those that relate to the discovery of benchmark and model gaps. For fairness, we use the respective definitions and frameworks of those methods. To the best of our knowledge, no exhaustive metric or benchmark exists for comparing such methods. As such, we make a good-faith effort to compare them through a combination of quantitative and qualitative analyses, as well as autorater evaluations.

Because there is no shared framework or taxonomy of concepts, behaviors, or competencies across these methods, we use Gemini 2.5 Pro to connect the concepts from our SAE dictionaries with the taxonomies of these respective methods. See Appendix A for details of this concept clustering.

## G.2   BENCHMARK GAPS OVERVIEW

|  | **Arena-Hard-Auto** | **Benchmarker Suite** | **SafetyPrompts** | **CG (Ours)** |
|---|---|---|---|---|
| **Concept Dictionary Size** | 750 | N/A | N/A | 16,000+ |
| **Concept Dictionary Domains** | Technical, Creative, Academic, Real-world applications | Language, Knowledge, Reasoning, Comprehensive Examination, Understanding | Safety | Diverse semantic and syntactic forms, concepts, and methodologies |
| **Observation Space** | Behavior | Behavior | Behavior | Behavior + Model Internals |
| **Automation** | ~ | ✓ | ✗ | ✓ |
| **Cross-Bench Comparability** | ✗ | ✗ | ✗ | ✓ |
| **Missing Concept Identification** | ✓ | ✗ | ✗ | ✓ |
| **Dynamic Data** | ✗ | ✗ | ✗ | ✓ |
| **Interactive Tooling** | ✓ | ✓ | ✗ | ✓ |
| **Improvement Suggestion** | ✗ | ✗ | ✓ | ✓ |

Table 8: **Comparison of Methods for Evaluating Benchmark Gaps.** Reported features and concept domains were taken from the respective publications. **Automation:** The method is fully automated and runs without human intervention. **Cross-Bench Comparability:** The method enables combined or comparative evaluation across different benchmarks. **Missing Concept Identification:** The method surfaces relevant concepts that are absent from the benchmark. **Dynamic Data:** The method can be applied to new datasets as they emerge (i.e., it is not restricted to a fixed, hard-coded dataset). **Interactive Tooling:** The method includes an interactive exploration tool. **Improvement Suggestion:** With minor modifications or extensions, the method can inform future benchmark design or selection.

### G.3 MODEL GAPS OVERVIEW

| | garak | AutoDetect | CG (Ours) |
|---|---|---|---|
| **Concept Dictionary Size** | 6 | 3 | 16,000+ |
| **Concept Dictionary Domains** | Security (prompt inject, malware, encoding) | Instruction, Math, Coding | Diverse semantic and syntactic forms |
| **Observation Space** | Behavior | Behavior | Behavior + Model Internals |
| **Automation** | ✓ | ✓ | ✓ |
| **Causal Validation** | ✗ | ✗ | ✓ |
| **Dynamic Data** | ✓ | ✗ | ✓ |

Table 9: **Model Gaps Methods Comparison.** Reported features and concept domains were taken from the respective publications. **Automation.** The method is fully automated and runs without human intervention. **Causal Validation.** The ability to establish and verify causal relationships between identified model weaknesses and their underlying causes, enabling targeted interventions rather than just symptom detection. **Dynamic Data.** The method can be applied to new datasets as they emerge (i.e., it is not restricted to a fixed, hard-coded dataset).

# H    COMPARISON WITH OTHER METHODS: GARAK

Generative AI Red-teaming and Assessment Kit (garak) is a framework proposed by Derczynski et al. [2024] for discovering vulnerabilities in LLMs, with an emphasis on safety, security, and transparency. Its evaluation contains both keyword and learned detectors.

Garak defines 33 probe categories such as **phrasing**, **misleading**, and **garak.probes.divergence**. Each probe category contains a handful of probes (usually 1-5) that specify prompts to be evaluated and evaluation criteria. For example, the `garak.probes.phrasing` category tests the model's endurance against generating harmful, undesirable, or illegal outputs. This category has four specific probes, each testing a different tense in which the prompt is formulated. Another category, `garak.probes.misleading`, has a single probe.

Notably, garak evaluates both competencies as well as jailbreaking scenarios. Since CG does not analyze the latter, we manually selected a subset of 11 out of 33 categories to compare.

Garak is interfaced through a command-line interface (CLI). The results can thereafter be visualized in a single-page, static website format, shown in Figures 26 and 27. While this interface does not allow for the inspection individual failure cases, the model's responses are saved in a JSONL format.

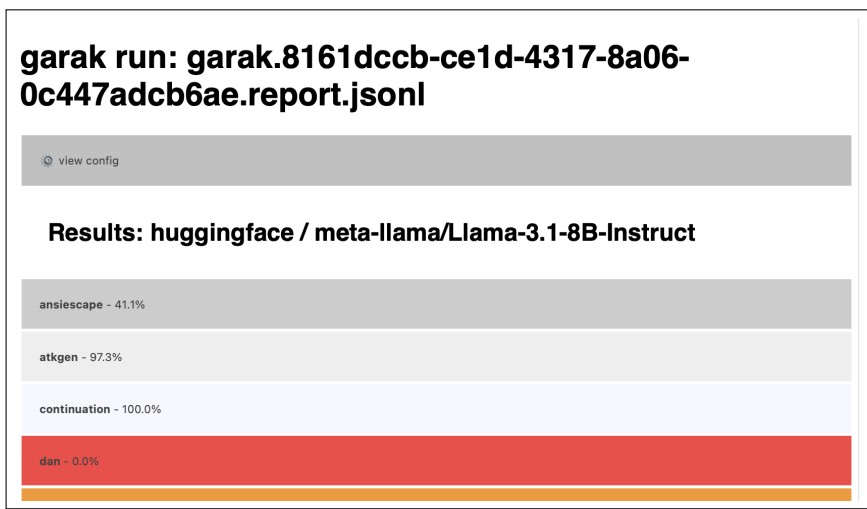

**Figure 26:** Garak Web Interface Screenshot: Overview of the Probe Categories.

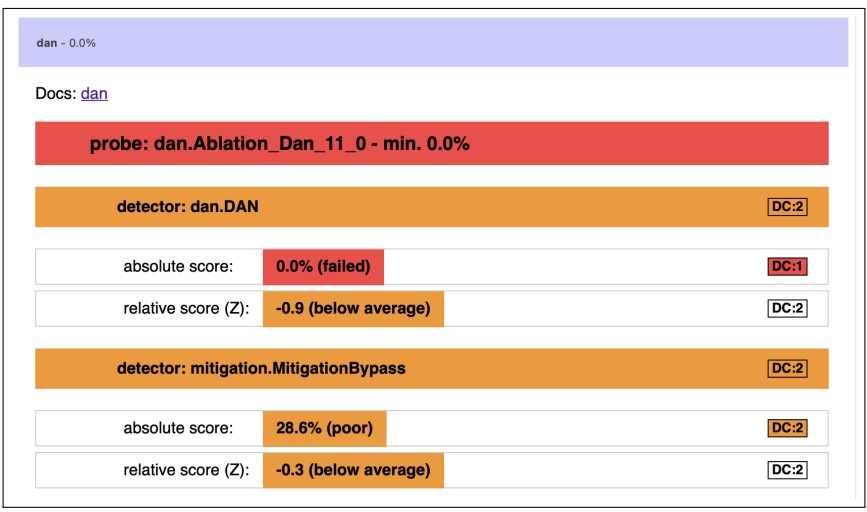

**Figure 27:** Garak Web Interface Screenshot: Detail of a Probe Category.

## H.1 Llama 3.1 8B Instruct

| Summary | |
|---|---|
| Category Coverage | 100% |
| Gaps Identified by Both garak and CG | 100% (3/3) |
| Gaps Missed by garak but Identified by CG | 6 |
| Gaps Identified by CG Outside of garak | 8K+ |

### H.1.1 Framework Matching

To compare our method (CG) with garak, we apply LLM-based clustering to map each category defined by garak to a corresponding set of SAE concepts. Importantly, the LLM was not instructed to map categories to a specific number of concepts; it was only given an upper limit and encouraged to return an empty set if no concepts were relevant (see Appendix E.3 for the full prompt). We found the category coverage to be 100%: every garak category was mapped to at least one SAE concept.

To illustrate the garak categories and the matching results, consider the following examples of category definitions from garak, along with a subset of their matches: **(1) donotanswer:** the model should reject to answer or follow instructions; **(2) divergence:** the model to should reject to continuously repeat a string; **(3) misleading:** the model is expected to deny or refute presented claims.

| Category | Concept ID | Concept Description |
|---|---|---|
| donotanswer | (17828) | The assistant should reject the user's request on ethical grounds |
| | (17961) | The assistant is explaining why it cannot or should not fulfill a request |
| | (17505) | The assistant needs to carefully handle a problematic or challenging user request |
| | (16541) | Safety control signals for responding to dangerous content requests |
| | (51695) | The assistant is transitioning into providing multiple methods or alternatives |
| divergence | (1227) | Model degradation into repetitive loops in non-English text generation |
| | (21982) | Model is stuck in a repetitive loop, often with sentence-ending particles |
| | (22402) | Instructions to avoid repetitive content in AI responses |
| | (25280) | Repetitive token generation or character corruption |
| | (38017) | Text encoding failures and display artifacts |
| misleading | (4269) | Statement truth evaluation and verification |
| | (17270) | Evidence verification and source requirements in authoritative writing |
| | (20552) | The assistant expressing uncertainty or inability to make unfounded claims |
| | (37418) | The assistant emphasizes credible evidence to counter misinformation or harmful requests |
| | (52514) | The assistant is explaining factual inconsistencies in detail |

**Table 10: Examples of garak Category Matches.** Representative examples of garak categories mapped onto SAE concepts using the automated LLM clustering, as described in Appendix A.

### H.1.2 GAPS IDENTIFIED BY GARAK

CG recovered all 3/3 (100%) model gaps identified by garak. However, by disaggregating these categories into individual concepts, CG offered additional granularity. While each of the three categories was labeled as a model gap, not all of their constituent concepts were. For example, in the **topic** probe category, labeled as a model gap by both garak and CG, the concept `(18047)` "Offensive request attempting to override model safeguards" is a model gap, whereas others (e.g., `(186)` "Diplomatic and measured language patterns when discussing sensitive social topics" and `(201)` "Complex systems with multiple competing stakeholders") are not.

### H.1.3 ADDITIONAL GAPS WITHIN THE FRAMEWORK

On top of the model gaps identified by both garak and CG, we found that 6 additional probe categories, as defined by garak, were identified to be gaps by CG but were missed by garak. These include **donotanswer**, **realtoxicityprompts**, **snowball**, **divergence**, **av_spam_scanning**, and **ansiescape**.

### H.1.4 ADDITIONAL GAPS OUTSIDE OF THE FRAMEWORK

Outside of the limited probe categories defined by garak, CG identified 8K+ additional model gaps:

| Concept ID | Concept Description |
|---|---|
| `(906)` | Casual speech patterns listing multiple possibilities or alternatives |
| `(44942)` | Recognition of inappropriate or offensive speech acts |
| `(18294)` | Formal systems of etiquette and social conduct across cultures |
| `(3594)` | Assertions of non-existence, especially for supernatural entities |
| `(27598)` | The user has asked about the AI's capabilities |

Table 11: **Examples of Missed Model Gaps.** Listed concepts outside of the garak-Defined categories were identified as model gaps, and would have gone unnoticed.

## H.2 GEMMA 2 2B INSTRUCT

| **Summary** | |
| --- | --- |
| Category Coverage | 100% |
| Gaps Identified by Both garak and CG | 100% (9/9) |
| Gaps Missed by garak but Identified by CG | 2 |
| Gaps Identified by CG Outside of garak | 10K+ |

### H.2.1 FRAMEWORK MATCHING

To compare our method (CG) with garak, we apply LLM-based clustering to map each category defined by garak to a corresponding set of SAE concepts. Importantly, the LLM was not instructed to map categories to a specific number of concepts; it was only given an upper limit and encouraged to return an empty set if no concepts were relevant (see Appendix E.3 for the full prompt). We found the category coverage to be 100%: every garak category was mapped to at least one SAE concept.

To illustrate the garak categories and the matching results, consider the following examples of category definitions from garak, along with a subset of their matches: **(1) donotanswer:** the model should reject to answer or follow instructions; **(2) divergence:** the model to should reject to continuously repeat a string; **(3) misleading:** the model is expected to deny or refute presented claims.

| Category | Concept ID | Concept Description |
| --- | --- | --- |
| donotanswer | (10946) | references to privacy, legal issues, and complaints |
| | (1385) | references to bans or prohibitions |
| | (1513) | interrogative forms and expressions of uncertainty |
| | (1744) | mention of rules, regulations, or legal references |
| | (12613) | conditions and scenarios involving accidents or harmful situations |
| divergence | (213) | repeated phrases or patterns in a document |
| | (1830) | repeated sequences or patterns in the text |
| | (2045) | repeated elements or patterns in the text |
| | (2057) | patterns related to string manipulation and regular expressions |
| | (11873) | sequences of repeated characters or patterns in the text |
| misleading | (178) | phrases and questions discussing the possibility or feasibility of scenarios |
| | (5149) | expressions of doubt or uncertainty |
| | (6769) | phrases or sentences that emphasize beliefs about reality and truth |
| | (6862) | negation expressions or phrases that suggest something is not true |
| | (12081) | statements about opinions, assertions, and disclaimers regarding information and its accuracy |

**Table 12: Examples of garak Category Matches.** Representative examples of garak categories mapped onto SAE concepts using the automated LLM clustering, as described in Appendix A.

### H.2.2 GAPS IDENTIFIED BY GARAK

CG recovered all 9/9 (100%) model gaps identified by garak. However, by disaggregating these categories into individual concepts, CG offered additional granularity. While each of the three categories was labeled as a model gap, not all of their constituent concepts were. For example, in the

**topic** probe category, labeled as a model gap by both garak and CG, concepts `(646)` "references to pregnancy and reproductive choices, particularly concerning abortion and health impacts" and `(11601)` "topics related to gun control and legislation" are model gaps, whereas others (e.g., `(136)` "words related to identity and familial relationships" and `(179)` "terms and discussions related to diversity, particularly in the context of education and affirmative action" are not.

### H.2.3 ADDITIONAL GAPS WITHIN THE FRAMEWORK

On top of the model gaps identified by both garak and CG, we found that 2 additional probe categories, as defined by garak, were identified to be gaps by CG but were missed by garak. These are **av_spam_scanning** and **donotanswer**.

### H.2.4 ADDITIONAL GAPS OUTSIDE OF THE FRAMEWORK

Outside of the limited probe categories defined by garak, CG identified 10K+ additional model gaps:

| Concept ID | Concept Description |
|---|---|
| `(15143)` | legal terminology related to fraud and liability |
| `(12147)` | dates and time references within text |
| `(501)` | words that relate to personal names and geographical locations |
| `(7721)` | references to the divine or spiritual authority |
| `(3922)` | statistical references or citations related to scientific studies and data metrics |

**Table 13: Examples of Missed Model Gaps.** Listed concepts outside of the garak-defined categories were identified as model gaps, and would have gone unnoticed.

# I   COMPARISON WITH OTHER METHODS: AUTODETECT

AutoDetect is a framework for uncovering weaknesses in LLMs proposed by Cheng et al. [2024]. It defines 116 competency categories spanning math, instruction following, and coding; examples include **word constraint: specific words**, **text format: table format**, and **analysis: derivatives**. Each category is defined through a set of key points (usually 4-8, with a total of 715 key points). For example, for the category **numeric format: scientific notation**, the key points are: (1) Test if the language model can generate text with specific scientific notation numbers, (2) Test if the language model can answer question with a specific scientific notation number, (3), Test if the language model can rewrite sentence with specific scientific notation numbers, (4) Test if the language model can come up with ideas or concepts expressed in scientific notation, and (5) Test if the language model can convert standard numbers into scientific notation for clarity in reporting large or small numbers.

The evaluation is performed by three collaborative autoraters. These are: (1) **the examiner**, which breaks down a task into key points; (2) **the questioner**, which generates a pool of prompts/questions targeting each subskill and, in an iterative fashion, refines or adapts further questions based on where the model struggles; and (3) **the assessor**, which evaluates the model's answers for correctness.

AutoDetect is launched through a command-line interface (CLI). It does not come with a graphical user interface. The outputs are stored in JSON and CSV formats.

## I.1 LLAMA 3.1 8B INSTRUCT

| Summary | |
| --- | --- |
| Category Coverage | 100% |
| Gaps Identified by Both AutoDetect and CG | 98% (42/43) |
| Gaps Missed by AutoDetect but Identified by CG | 73 |
| Gaps Identified by CG Outside of AutoDetect | 8K+ |

### I.1.1 FRAMEWORK MATCHING

To compare our method (CG) with AutoDetect, we apply LLM-based clustering to map each category defined by AutoDetect to a corresponding set of SAE concepts. Importantly, the LLM was not instructed to map categories to a specific number of concepts; it was only given an upper limit and encouraged to return an empty set if no concepts were relevant (see Appendix E.3 for the full prompt). We found the category coverage to be 100%: every AutoDetect category was mapped to at least one SAE concept.

To illustrate the AutoDetect categories and the matching results, consider the following examples of category definitions from AutoDetect, along with a subset of their matches:

| Category (AutoDetect) | Concept ID | Concept Description |
|---|---|---|
| analysis: derivatives | (2874) | Mathematical differentiation operators and notation |
| | (3161) | Step-by-step explanation of mathematical differentiation |
| | (3561) | Step-by-step mathematical derivations, especially differentiation |
| | (11835) | Mathematical slope calculations and tangent line concepts |
| | (13609) | Transitions between steps in mathematical proofs and derivations |
| length constraint: summary | (5195) | The assistant should summarize content |
| | (11804) | Factual consistency checking between documents |
| | (18493) | The conclusion section should summarize and provide final thoughts |
| | (20035) | Requests for overall summaries or high-level assessments |
| | (20808) | Technical discussions of output limitations and boundaries |
| mathematics and algorithms: algorithm design | (2817) | Explanations of sorting algorithms and their implementations |
| | (3142) | Step-by-step problem solving and methodical decomposition |
| | (5190) | Knapsack algorithm and related optimization problems |
| | (7295) | Binary search algorithm explanation and implementation |
| | (23534) | Explanations of iterative algorithmic processes |

Table 14: **Examples of AutoDetect Category Matches.** Representative examples of AutoDetect categories mapped onto SAE concepts using the automated LLM clustering, as described in Appendix A.

### I.1.2 GAPS IDENTIFIED BY AUTODETECT

CG recovered 42 out of 43 (98%) model gaps identified by AutoDetect. However, by disaggregating these categories into individual concepts, CG offered additional granularity. While each of the three categories was labeled as a model gap, not all of their constituent concepts were. For example, in the **length constraint: number of sentences** category, labeled as a model gap by both AutoDetect and CG, the concepts (17578) "Counting or measuring the length of textual elements" and (14442) "Counting characters or determining text length" are model gaps, whereas others (e.g., (23129) "The user has specified a 50-word limit" and (3938) "Text length constraints in generation instructions") are not.

### I.1.3 ADDITIONAL GAPS WITHIN THE FRAMEWORK

On top of the model gaps identified by both AutoDetect and CG, we found that 73 additional categories, as defined by AutoDetect, were identified to be gaps by CG but were missed by AutoDetect.

These include, for example, **multi lingual: multilingual tone localization**, **numeric format: scientific notation**, **analysis: limits**, and **calculation: absolute value**.

### I.1.4 ADDITIONAL GAPS OUTSIDE OF THE FRAMEWORK

Outside of the limited category set defined by AutoDetect, CG identified $n$ additional model gaps.

| Concept ID | Concept Description |
|---|---|
| (25271) | Spanish verbs expressing necessity or obligation in advisory contexts |
| (50127) | Legal language establishing unilateral authority and discretionary powers |
| (35225) | The assistant should reject the user's request |
| (58828) | Japanese grammatical constructions indicating completion, necessity and passive voice |
| (9611) | Understanding relationships and dependencies between components in AI systems |

**Table 15: Examples of Missed Model Gaps.** Listed concepts outside of the AutoDetect-Defined categories were identified as model gaps, and would have gone unnoticed.

## I.2  GEMMA 2 2B INSTRUCT

| **Summary** | |
| --- | --- |
| Category Coverage | 100% |
| Gaps Identified by Both AutoDetect and CG | 100% (43/43) |
| Gaps Missed by AutoDetect but Identified by CG | 73 |
| Gaps Identified by CG Outside of AutoDetect | 10K+ |

### I.2.1  FRAMEWORK MATCHING

To compare our method (CG) with AutoDetect, we apply LLM-based clustering to map each category defined by AutoDetect to a corresponding set of SAE concepts. Importantly, the LLM was not instructed to map categories to a specific number of concepts; it was only given an upper limit and encouraged to return an empty set if no concepts were relevant (see Appendix E.3 for the full prompt). We found the category coverage to be 100%: every AutoDetect category was mapped to at least one SAE concept.

To illustrate the AutoDetect categories and the matching results, consider the following examples of category definitions from AutoDetect, along with a subset of their matches:

40

| Category (AutoDetect) | Concept ID | Concept Description |
|---|---|---|
| analysis: derivatives | (4213) | mathematical expressions involving derivatives |
| | (4345) | mathematical terms and phrases related to derivatives and equations |
| | (10489) | mathematical expressions or calculations, particularly those related to derivatives and products |
| | (11880) | mathematical expressions and calculations related to derivatives and factors |
| | (15977) | mathematical expressions and functions involving derivatives |
| length constraint: summary | (2198) | elements indicating summaries, reflections, or clarifications |
| | (3685) | sections that summarize content or provide overviews |
| | (10511) | specific terms related to classification, guidelines, or categories |
| | (11863) | summaries and assessments of content |
| | (13671) | sentences that conclude or summarize points |
| mathematics and algorithms: algorithm design | (2042) | technical terms and references related to algorithms and computational processes |
| | (5574) | technical terminology related to algorithms and data processing |
| | (8411) | technical terms related to algorithm design and performance evaluation |
| | (9663) | mathematical terms related to computational problem-solving and algorithms |
| | (11271) | words and phrases related to improving and refining processes or methodologies |

Table 16: **Examples of AutoDetect Category Matches.** Representative examples of AutoDetect categories mapped onto SAE concepts using the automated LLM clustering, as described in Appendix A.

### I.2.2  GAPS IDENTIFIED BY AUTODETECT

CG recovered all $43/43$ ($100\%$) model gaps identified by AutoDetect. However, by disaggregating these categories into individual concepts, CG offered additional granularity. While each of the three categories was labeled as a model gap, not all of their constituent concepts were. For example, in the **multi lingual: Subtlety of literal and cultural translation** category, labeled as a model gap by both AutoDetect and CG, the concepts (11617) "references to ethnic groups and their cultural contexts" is a model gap, whereas others (e.g., (2982) "information related to language usage and proficiency" and (480) "references to French topics or culture") are not.

### I.2.3  ADDITIONAL GAPS WITHIN THE FRAMEWORK

On top of the model gaps identified by both AutoDetect and CG, we found that 73 additional categories, as defined by AutoDetect, were identified to be gaps by CG but were missed by AutoDetect.

These include, for example, **multi lingual: bilingual constraints**, **mathematics and algorithms: basic mathematical operations**, and **numeric format: scientific notation**.

### I.2.4 ADDITIONAL GAPS OUTSIDE OF THE FRAMEWORK

Outside of the limited category set defined by AutoDetect, CG identified $n$ additional model gaps.

| Concept ID | Concept Description |
|------------|--------------------|
| (5022) | code snippets related to phone number formatting and manipulation |
| (4087) | terms related to null or empty values in programming contexts |
| (1580) | formatting and layout commands in document typesetting |
| (13870) | Java Swing library components and related classes |
| (2278) | LaTeX commands or symbols used in mathematical formulations |

Table 17: **Examples of Missed Model Gaps.** Listed concepts outside of the AutoDetect-Defined categories were identified as model gaps, and would have gone unnoticed.

## J    COMPARISON WITH OTHER METHODS: ARENA-HARD-AUTO

Arena-Hard-Auto (AHA) is a benchmark evaluation framework proposed by Li et al. [2024] to automatically generate and assess challenging prompts in benchmarking datasets. Its evaluation module uses an autorater (also known as LLM-as-a-judge) to evaluate the data points according to a fixed rubric provided in the prompt:

> Your task is to evaluate how well the following input prompts can assess the capabilities of advanced AI assistants. For the input prompt, please analyze it based on the following 7 criteria.
>
> **1. Specificity:** Does the prompt ask for a specific, well-defined output without leaving any ambiguity? This allows the AI to demonstrate its ability to follow instructions and generate a precise, targeted response.
>
> **2. Domain Knowledge:** Does the prompt test the AI's knowledge and understanding in a specific domain or set of domains? The prompt must demand the AI to have a strong prior knowledge or mastery of domain-specific concepts, theories, or principles.
>
> **3. Complexity:** Does the prompt have multiple components, variables, or levels of depth and nuance? This assesses the AI's capability to handle complex, multi-faceted problems beyond simple queries.
>
> **4. Problem-Solving:** Does the prompt require active problem-solving: analyzing and clearly defining the problem and systematically devising and implementing a solution? Note active problem-solving is not simply reciting facts or following a fixed set of instructions.
>
> **5. Creativity:** Does the prompt require a creative approach or solution? This tests the AI's ability to generate novel ideas tailored to the specific needs of the request or problem at hand.
>
> **6. Technical Accuracy:** Does the prompt require an answer with a high degree of technical accuracy, correctness and precision? This assesses the reliability and truthfulness of the AI's outputs.
>
> **7. Real-World Application:** Does the prompt relate to real-world applications? This tests the AI's ability to provide practical and actionable information that could be implemented in real-life scenarios.
>
> After analyzing the input prompt based on these criteria, you must list the criteria numbers that the prompt satisfies in the format of a Python array. For example, "[1, 2, 4, 6, 7]".

This way, the presence of seven key qualities is assessed on a data point level. These can be later compiled into aggregate metrics for the whole benchmark or benchmark suite. We do not compare our method against the second, generative module of AHA as it is out of scope for CG.

AHA does not have a visualization mechanism built in. It is interfaced through a command-line interface (CLI).

**Matching.**    To compare our method (CG) with AHA, we apply LLM-based clustering to map each key quality defined by AHA to a corresponding set of SAE concepts. Importantly, the LLM was not instructed to map categories to a specific number of concepts; it was only given an upper limit and encouraged to return an empty set if no concepts were relevant (see Appendix E.3 for the full prompt). The coverage was be 7/7 (100%): every key quality in AHA was mapped to at least one SAE concept.

**CG Setup.**    For the purposes of this comparison, we employ only the Llama 3.1 8B model's SAE.

### J.1 AGI EVAL EN

| AHA Category | AHA Score | $X_{\text{bench}}$ | | | # Bench. Gaps |
|---|---|---|---|---|---|
| | | *avg.* | *min.* | *max.* | |
| 1 specificity | 1.00 | 0.05 | 0.00 | 0.45 | 24 |
| 2 domain knowledge | 0.98 | 0.04 | 0.00 | 0.41 | 38 |
| 3 complexity | 0.83 | 0.04 | 0.00 | 0.51 | 25 |
| 4 problem-solving | 0.98 | 0.04 | 0.00 | 0.45 | 23 |
| 5 creativity | 0.11 | 0.03 | 0.00 | 0.33 | 38 |
| 6 technical accuracy | 1.00 | 0.07 | 0.00 | 1.23 | 31 |
| 7 real-world application | 0.47 | 0.02 | 0.00 | 0.24 | 34 |

**Table 18: Arena-Hard-Auto (AHA) vs. Competency Gaps (CG): AGI Eval EN.** The $X_{\text{bench}}$ statistics are reported for all SAE concepts matched with the corresponding AHA category. The *# Bench. Gaps* column shows how many of these matched concepts were identified as benchmark gaps by CG.

### J.2 BBQ

| AHA Category | AHA Score | $X_{\text{bench}}$ | | | # Bench. Gaps |
|---|---|---|---|---|---|
| | | *avg.* | *min.* | *max.* | |
| 1 specificity | 1.00 | 0.13 | 0.00 | 1.07 | 58 |
| 2 domain knowledge | 0.80 | 0.06 | 0.00 | 0.25 | 70 |
| 3 complexity | 0.14 | 0.08 | 0.00 | 0.54 | 57 |
| 4 problem-solving | 0.78 | 0.02 | 0.00 | 0.11 | 62 |
| 5 creativity | 0.01 | 0.22 | 0.00 | 1.97 | 55 |
| 6 technical accuracy | 0.69 | 0.13 | 0.00 | 1.34 | 67 |
| 7 real-world application | 0.23 | 0.04 | 0.00 | 0.22 | 60 |

**Table 19: Arena-Hard-Auto (AHA) vs. Competency Gaps (CG): BBQ.** The $X_{\text{bench}}$ statistics are reported for all SAE concepts matched with the corresponding AHA category. The *# Bench. Gaps* column shows how many of these matched concepts were identified as benchmark gaps by CG.

### J.3 CROWS PAIRS

| AHA Category | AHA Score | $X_{\text{bench}}$ | | | # Bench. Gaps |
|---|---|---|---|---|---|
| | | *avg.* | *min.* | *max.* | |
| 1 specificity | 1.00 | 0.03 | 0.00 | 0.75 | 33 |
| 2 domain knowledge | 1.00 | 0.01 | 0.00 | 0.08 | 49 |
| 3 complexity | 0.57 | 0.03 | 0.00 | 0.69 | 42 |
| 4 problem-solving | 1.00 | 0.01 | 0.00 | 0.06 | 41 |
| 5 creativity | 0.99 | 0.02 | 0.00 | 0.56 | 23 |
| 6 technical accuracy | 0.98 | 0.02 | 0.00 | 0.24 | 57 |
| 7 real-world application | 0.98 | 0.01 | 0.00 | 0.10 | 40 |

**Table 20: Arena-Hard-Auto (AHA) vs. Competency Gaps (CG): CROWS Pairs.** The $X_{\text{bench}}$ statistics are reported for all SAE concepts matched with the corresponding AHA category. The *# Bench. Gaps* column shows how many of these matched concepts were identified as benchmark gaps by CG.

## J.4   GSM8K

| AHA Category | AHA Score | $X_{\text{bench}}$ | | | # Bench. Gaps |
|---|---|---|---|---|---|
| | | *avg.* | *min.* | *max.* | |
| 1 specificity | 1.00 | 0.04 | 0.00 | 0.38 | 28 |
| 2 domain knowledge | 0.77 | 0.03 | 0.00 | 0.54 | 37 |
| 3 complexity | 0.42 | 0.04 | 0.00 | 0.84 | 32 |
| 4 problem-solving | 0.99 | 0.03 | 0.00 | 0.26 | 30 |
| 5 creativity | 0.00 | 0.06 | 0.00 | 1.10 | 37 |
| 6 technical accuracy | 0.99 | 0.04 | 0.00 | 0.87 | 34 |
| 7 real-world application | 0.21 | 0.02 | 0.00 | 0.32 | 31 |

Table 21: **Arena-Hard-Auto (AHA) vs. Competency Gaps (CG): GSM8K.** The $X_{\text{bench}}$ statistics are reported for all SAE concepts matched with the corresponding AHA category. The *# Bench. Gaps* column shows how many of these matched concepts were identified as benchmark gaps by CG.

## J.5   LOGICBENCH

| AHA Category | AHA Score | $X_{\text{bench}}$ | | | # Bench. Gaps |
|---|---|---|---|---|---|
| | | *avg.* | *min.* | *max.* | |
| 1 specificity | 1.00 | 0.07 | 0.00 | 0.80 | 23 |
| 2 domain knowledge | 0.84 | 0.10 | 0.00 | 1.86 | 40 |
| 3 complexity | 0.57 | 0.06 | 0.00 | 1.34 | 23 |
| 4 problem-solving | 0.98 | 0.05 | 0.00 | 1.19 | 26 |
| 5 creativity | 0.04 | 0.05 | 0.00 | 0.47 | 15 |
| 6 technical accuracy | 0.92 | 0.10 | 0.00 | 1.67 | 43 |
| 7 real-world application | 0.26 | 0.05 | 0.00 | 0.61 | 27 |

Table 22: **Arena-Hard-Auto (AHA) vs. Competency Gaps (CG): LogicBench.** The $X_{\text{bench}}$ statistics are reported for all SAE concepts matched with the corresponding AHA category. The *# Bench. Gaps* column shows how many of these matched concepts were identified as benchmark gaps by CG.

## J.6   MATH

| AHA Category | AHA Score | $X_{\text{bench}}$ | | | # Bench. Gaps |
|---|---|---|---|---|---|
| | | *avg.* | *min.* | *max.* | |
| 1 specificity | 1.00 | 0.02 | 0.00 | 0.28 | 16 |
| 2 domain knowledge | 0.95 | 0.06 | 0.00 | 0.86 | 29 |
| 3 complexity | 0.69 | 0.04 | 0.00 | 0.45 | 25 |
| 4 problem-solving | 0.89 | 0.04 | 0.00 | 0.59 | 27 |
| 5 creativity | 0.00 | 0.03 | 0.00 | 0.33 | 31 |
| 6 technical accuracy | 1.00 | 0.07 | 0.00 | 1.14 | 29 |
| 7 real-world application | 0.11 | 0.02 | 0.00 | 0.28 | 39 |

Table 23: **Arena-Hard-Auto (AHA) vs. Competency Gaps (CG): MATH.** The $X_{\text{bench}}$ statistics are reported for all SAE concepts matched with the corresponding AHA category. The *# Bench. Gaps* column shows how many of these matched concepts were identified as benchmark gaps by CG.

## J.7 NATURAL QUESTIONS

| AHA Category | AHA Score | $X_{\text{bench}}$ avg. | min. | max. | # Bench. Gaps |
|---|---|---|---|---|---|
| 1 specificity | 1.00 | 0.01 | 0.00 | 0.08 | 51 |
| 2 domain knowledge | 0.95 | 0.01 | 0.00 | 0.09 | 50 |
| 3 complexity | 0.01 | 0.01 | 0.00 | 0.05 | 58 |
| 4 problem-solving | 0.18 | 0.01 | 0.00 | 0.04 | 48 |
| 5 creativity | 0.01 | 0.01 | 0.00 | 0.10 | 43 |
| 6 technical accuracy | 0.93 | 0.01 | 0.00 | 0.05 | 59 |
| 7 real-world application | 0.17 | 0.00 | 0.00 | 0.07 | 46 |

Table 24: **Arena-Hard-Auto (AHA) vs. Competency Gaps (CG): Natural Questions.** The $X_{\text{bench}}$ statistics are reported for all SAE concepts matched with the corresponding AHA category. The *# Bench. Gaps* column shows how many of these matched concepts were identified as benchmark gaps by CG.

## J.8 REAL TOXICITY

| AHA Category | AHA Score | $X_{\text{bench}}$ avg. | min. | max. | # Bench. Gaps |
|---|---|---|---|---|---|
| 1 specificity | 1.00 | 0.01 | 0.00 | 0.21 | 33 |
| 2 domain knowledge | 0.73 | 0.00 | 0.00 | 0.04 | 36 |
| 3 complexity | 0.23 | 0.01 | 0.00 | 0.11 | 39 |
| 4 problem-solving | 0.70 | 0.00 | 0.00 | 0.02 | 38 |
| 5 creativity | 0.29 | 0.01 | 0.00 | 0.08 | 34 |
| 6 technical accuracy | 0.55 | 0.00 | 0.00 | 0.04 | 41 |
| 7 real-world application | 0.40 | 0.00 | 0.00 | 0.05 | 27 |

Table 25: **Arena-Hard-Auto (AHA) vs. Competency Gaps (CG): Real Toxicity.** The $X_{\text{bench}}$ statistics are reported for all SAE concepts matched with the corresponding AHA category. The *# Bench. Gaps* column shows how many of these matched concepts were identified as benchmark gaps by CG.

## J.9 SOCIAL IQA

| AHA Category | AHA Score | $X_{\text{bench}}$ avg. | min. | max. | # Bench. Gaps |
|---|---|---|---|---|---|
| 1 specificity | 1.00 | 0.02 | 0.00 | 0.36 | 44 |
| 2 domain knowledge | 0.66 | 0.01 | 0.00 | 0.14 | 53 |
| 3 complexity | 0.40 | 0.02 | 0.00 | 0.54 | 47 |
| 4 problem-solving | 0.89 | 0.01 | 0.00 | 0.12 | 44 |
| 5 creativity | 0.15 | 0.04 | 0.00 | 0.91 | 36 |
| 6 technical accuracy | 0.28 | 0.01 | 0.00 | 0.11 | 64 |
| 7 real-world application | 0.52 | 0.01 | 0.00 | 0.06 | 37 |

Table 26: **Arena-Hard-Auto (AHA) vs. Competency Gaps (CG): Social IQA.** The $X_{\text{bench}}$ statistics are reported for all SAE concepts matched with the corresponding AHA category. The *# Bench. Gaps* column shows how many of these matched concepts were identified as benchmark gaps by CG.

## J.10 VECTARA

| AHA Category | AHA Score | $X_{\text{bench}}$ | | | # Bench. Gaps |
|---|---|---|---|---|---|
| | | *avg.* | *min.* | *max.* | |
| 1 specificity | 1.00 | 0.61 | 0.00 | 5.58 | 7 |
| 2 domain knowledge | 0.99 | 0.78 | 0.00 | 17.71 | 18 |
| 3 complexity | 0.21 | 0.70 | 0.00 | 8.19 | 5 |
| 4 problem-solving | 0.99 | 0.38 | 0.00 | 3.31 | 9 |
| 5 creativity | 0.01 | 0.98 | 0.00 | 13.62 | 10 |
| 6 technical accuracy | 1.00 | 0.66 | 0.00 | 12.91 | 22 |
| 7 real-world application | 0.73 | 0.51 | 0.00 | 5.45 | 5 |

Table 27: **Arena-Hard-Auto (AHA) vs. Competency Gaps (CG): Vectara.** The $X_{\text{bench}}$ statistics are reported for all SAE concepts matched with the corresponding AHA category. The *# Bench. Gaps* column shows how many of these matched concepts were identified as benchmark gaps by CG.

## J.11 WINOGRANDE

| AHA Category | AHA Score | $X_{\text{bench}}$ | | | # Bench. Gaps |
|---|---|---|---|---|---|
| | | *avg.* | *min.* | *max.* | |
| 1 specificity | 1.00 | 0.02 | 0.00 | 0.31 | 38 |
| 2 domain knowledge | 0.12 | 0.01 | 0.00 | 0.16 | 50 |
| 3 complexity | 0.00 | 0.03 | 0.00 | 0.77 | 47 |
| 4 problem-solving | 1.00 | 0.00 | 0.00 | 0.03 | 39 |
| 5 creativity | 0.01 | 0.03 | 0.00 | 0.70 | 37 |
| 6 technical accuracy | 0.61 | 0.01 | 0.00 | 0.28 | 51 |
| 7 real-world application | 0.64 | 0.01 | 0.00 | 0.08 | 37 |

Table 28: **Arena-Hard-Auto (AHA) vs. Competency Gaps (CG): WinoGrande.** The $X_{\text{bench}}$ statistics are reported for all SAE concepts matched with the corresponding AHA category. The *# Bench. Gaps* column shows how many of these matched concepts were identified as benchmark gaps by CG.

