# OpenReview forum: "Uncovering Competency Gaps in Large Language Models and Their Benchmarks"
_ICLR.cc/2026/Conference — Submitted to ICLR 2026_

### Official Review · Reviewer_x8Ry · 2025-10-18

**Soundness:** 3
**Presentation:** 3
**Contribution:** 3
**Rating:** 6
**Confidence:** 4

**Summary:**

This paper presents a novel and unified methodology for (a) evaluating the capabilities of LLMs and (b) appraising the effectiveness of static Question-Answer benchmarks for testing for those capabilities. The authors use Sparse Auto-Encoders (SAEs), a technique from mechanistic interpretability, to do this.

Using (I assume) pretrained SAEs with a pre-defined dictionary of tens of thousands of concepts, the normalized activation for each concept for each benchmark item is computed by summing the activation for that node of the hidden layer for each token, divided by the number of tokens in the benchmark prompt. For each benchmark, its coverage of a concept is computed as the normalized activation divided by the average concept activation across all concepts and items in the benchmark. It is not totally clear to me, but I assume that this is done relative to one LLM, rather than both or all possible LLMs, a point I come back to later. The coverage across all benchmarks is the average coverage across the set of benchmarks. A similar strategy is taken for models, except the per-benchmark model performance is modulated by some 'performance scoring policy', which I take to be the model's actual (or average?) score on a particular item of a benchmark.

Using these metrics, the authors evaluate two models (Llama-3.2-8B and Gemma2-2B) on 10 benchmarks. They find that these benchmarks show skewed coverage across the concept space, missing important concepts like meta-cognition and legal concepts. Both models also perform poorly on important safety-related concepts, such as politely rejecting inappropriate requests.

**Strengths:**

This paper is commendable, and constitutes a novel and innovative contribution to the growing literature on AI Evaluation. It is reasonably clearly written and well presented, apart from a few parts highlighted below. The web app is a cool feature that is certainly a great contribution of the work, if the empirical results stand up. The results that common benchmarks fail to measure capabilities/concepts that they purport to is well-made and empirically backed up, in contrast to much of the theoretical and/or purely verbal arguments made on this topic (Burden, 2024; Hernández-Orallo, 2017, 2020; Jo and Wilson, 2025; Raji et al., 2021). The results that models fail on many safety-related concepts is a striking result worth emphasising, although this may be a feature of model size and/or degree of safety post-training.



Burden, J. (2024). Evaluating ai evaluation: Perils and prospects. arXiv preprint arXiv:2407.09221.
Hernández-Orallo, J. (2020). AI evaluation: On broken yardsticks and measurement scales. In Workshop on evaluating evaluation of AI systems at AAAI. Menlo Park: Association for the Advancement of Artificial Intelligence.
Hernández-Orallo, J. (2017). Evaluation in artificial intelligence: from task-oriented to ability-oriented measurement. Artificial Intelligence Review, 48(3), 397-447.
Jo, N., & Wilson, A. (2025). What Does Your Benchmark Really Measure? A Framework for Robust Inference of AI Capabilities. arXiv preprint arXiv:2509.19590.
Raji, I. D., Bender, E. M., Paullada, A., Denton, E., & Hanna, A. (2021). AI and the everything in the whole wide world benchmark. arXiv preprint arXiv:2111.15366.

**Weaknesses:**

The paper currently has three weaknesses, in decreasing order of severity:
1. There are some reliability and validity checks to be done. First, it would be useful to verify that the same (or broadly the same) benchmark gaps are recovered with (a) different language models doing the prediction (or even just different seeds/temperatures) and (b) perturbed benchmark data (i.e., syntactic/superficial variants of the same prompts). This would verify that the benchmark gaps are reliable and not simply statistical artefacts of the current set up. Second, checking the predictive validity of the model gaps is imperative. Figure 5 is an anecdotal example, but it is necessary to find a set of model gaps on a subset of the benchmarks and then test the models on items from the held out set of benchmarks that correspond to those concepts. This would show that the model gaps are predictive of performance on new, unseen data. The authors could report in a table the success rates on held-out items from model-gap and non-model-gap concepts, so that the difference is clear.
2. The result about models performing poorly when, for instance, rejecting certain requests from users, is intriguing. The authors should consider running a parallel analysis with Llama-Guard-3-8B to see if that model gap disappears with post-training. If so, this would be great evidence in favour of the conclusion that Llama-3.1-8B genuinely has a safety gap.
3. There is some missing literature. Burden (2024), Burden et al. (2023), and Hernández-Orallo (2017, 2020) all discuss the problems of aggregating scores on benchmarks which is discussed in the first paragraph of the introduction. Similarly, much recent work has explored the redundancy of common NLP benchmarks and propose sophisticated strategies to determine and overcome that redundancy: Raji et al. (2021), Kipnis et al. (2024), Wang et al. (2025), Polo et al. (2024), Zhou et al. (2025). Indeed, the related work section should discuss Item Response Theory (IRT) as one alternative method for doing data-driven 'model gap'/capability discovery in large language models, which contrasts with the current approach of using predefined concept dictionaries (which is much more in-line with Burden et al. 2023).
4. There are a number of textual problems with the paper at the moment. The most important are handled by the first two of my questions below, namely, how benchmark gaps are computed and how the performance scoring policy is defined. I also think a similar box to the 'Model Gap' box in Section 3.2 is necessary for Benchmark Gap in Section 3.1, for clarity and consistency. There is a syntax error on lines 174-5, it should be: "that quantifies the degree to which a concept c was activated...". There is a missing ref to the appendix on line 332. I would recommend a read-through to check all syntax and references.


Burden, J. (2024). Evaluating ai evaluation: Perils and prospects. arXiv preprint arXiv:2407.09221.
Burden, J., Voudouris, K., Burnell, R., Rutar, D., Cheke, L., & Hernández-Orallo, J. (2023). Inferring capabilities from task performance with bayesian triangulation. arXiv preprint arXiv:2309.11975.
Hernández-Orallo, J. (2020). AI evaluation: On broken yardsticks and measurement scales. In Workshop on evaluating evaluation of AI systems at AAAI. Menlo Park: Association for the Advancement of Artificial Intelligence.
Hernández-Orallo, J. (2017). Evaluation in artificial intelligence: from task-oriented to ability-oriented measurement. Artificial Intelligence Review, 48(3), 397-447.
Kipnis, A., Voudouris, K., Buschoff, L. M. S., & Schulz, E. (2024). metabench--A Sparse Benchmark of Reasoning and Knowledge in Large Language Models. arXiv preprint arXiv:2407.12844.
Polo, F. M., Weber, L., Choshen, L., Sun, Y., Xu, G., & Yurochkin, M. (2024). tinyBenchmarks: evaluating LLMs with fewer examples. arXiv preprint arXiv:2402.14992.
Raji, I. D., Bender, E. M., Paullada, A., Denton, E., & Hanna, A. (2021). AI and the everything in the whole wide world benchmark. arXiv preprint arXiv:2111.15366.
Wang, Y., Ying, J., Cao, Y., Ma, Y., & Jiang, Y. (2025). EffiEval: Efficient and Generalizable Model Evaluation via Capability Coverage Maximization. arXiv preprint arXiv:2508.09662.
Zhou, L., Pacchiardi, L., Martínez-Plumed, F., Collins, K. M., Moros-Daval, Y., Zhang, S., ... & Hernández-Orallo, J. (2025). General scales unlock ai evaluation with explanatory and predictive power. arXiv preprint arXiv:2503.06378.

**Questions:**

* It is not clear to me how benchmark gaps are computed. Is it with respect to one or both of the language models? Should we interpret these coverage scores more as expectations over the space of possible language models?
* How is the performance scoring policy for the model gaps computed? I was assuming it was related to the score of the model on that benchmark item, but that would be a binary pass or fail, right? Or is it the log-probability of the correct answer token, the proportion of correct answers over some set of trials with high temperature, or something else entirely? It should be described somewhere.
* How does the benchmark gap calculation work if neither model is very good at, say, meta-cognition or legal reasoning? Surely this could be a reason why those concepts are misrepresented - i.e., simply because the models with respect to whom the benchmark gap scores are computed are incapable of performing well on those tasks. I'm sure I missed something there.
* What is the base rate activation of each of the concepts in each SAE. Is it possible that some of the low activating concepts simply do not ever see the high rates of activation of the others? How do you control for this base rate effect?

---

> ### Author Response · Authors · 2025-11-22
>
> We thank the reviewer for their very helpful feedback and for highlighting the "novel and innovative contribution" of our work. We are glad the reviewer found the web application to be a "cool feature" and appreciated that our results concerning benchmark failures and safety concepts were empirically backed.
>
> Below, we address the suggestions and questions raised. Revisions in the manuscript are tracked in red.
>
> ---
>
> **> C1. Reliability and validity checks.** We agree that establishing robustness is imperative. Following your suggestions, we have conducted extensive validation experiments, which are now detailed in Section 5.3: Robustness:
>
> - **Cross-Model/Cross-SAE Verification.** To verify that benchmark gaps are reliable and not artifacts of a specific model, we conducted a Cross-SAE analysis. We analyzed Llama 3.1 8B’s activations using Gemma 2 2B’s SAE, and vice versa. As shown in Table 3 and Figures 3 & 4, the distributions of gaps and performance scores remained consistent (e.g., safety concepts like "refusal" remained low-performing; "coding" remained high-performing).
> - **Perturbation Stability.** We re-ran the full analysis 100 times, randomly dropping 20% of examples per benchmark each time. The resulting standard deviations were extremely low (0.014 for $X_{model}$ and 0.025 for $X_{bench}$), confirming stability.
> - **Predictive Validity.** We performed an adversarial perturbation test where we identified the top 100 best- and worst-performing concepts and removed the data points most strongly associated with them. As predicted, removing "good" concept data lowered the global score, and removing "bad" concept data raised it (Section 5.3).
>
> We hope that the reliability and validity of the method is now better established.
>
> ---
>
> **> C2. Parallel analysis with Llama-Guard-3-8B.** This is an excellent suggestion. Due to the scope of the current rebuttal, we were unfortunately not able to perform this analysis currently, but we agree this would be a powerful validation of the tool’s utility, and we hope to perform this analysis as future work.
>
> ---
>
> **> C3. Missing literature.** Thank you for the very helpful suggestions on citations! We have updated Section 1: Introduction and Section 2: Related Work to include these references. We specifically expanded the discussion to contrast our dictionary-based approach with Item Response Theory and cited the works on benchmark redundancy (Raji et al., Kipnis et al., etc.) to better contextualize our contribution.
>
> ---
>
> **> C4. Textual suggestions.** We very much appreciate the time and care taken for the detailed feedback here. We have corrected the syntax errors (lines 174-5) and the missing appendix reference. Regarding the clarity of definitions, we have refined Section 3: Method to more clearly define the scoring policies. To improve the structural clarity requested, we added a new Figure 2 with recommended workflows, which acts as a visual guide for the methodology, complementing the text definitions. Please let us know if you believe that further clarification is needed.
>
> ---
>
> Please find the answers to the additional questions posed by the reviewer in the next comment.
>
> ---
>
> We hope these revisions and clarifications address your concerns, and we would be eager to collaborate further if you have additional suggestions to improve the paper's accessibility and rigor.

---

> ### Author Response · Authors · 2025-11-22
>
> This response continues the discussion posted in our previous response to this review.
>
> ---
>
> **> Q1. How are benchmark gaps computed?** Benchmark gaps are computed with respect to a particular model's activation patterns, and we believe that this is one strength of the approach – it allows us to evaluate models based on the models' own interpretations and activations, conditioned on inputs (e.g. benchmark examples). This is because the metric relies on $s_{c,i}$ (the activation of concept $c$ by model $M$ on input $i$). However, our Cross-SAE experiments (Section 5.3) suggest that these gaps are largely stable across models. Because different models tend to activate semantically similar features in response to the same text, a "benchmark gap" identified by Llama is highly likely to be a gap for Gemma as well.
>
> ---
>
> **> Q2. How is the performance scoring policy for the model gaps computed?** We have clarified this in Section 3.2. The scoring policy $m_b(i) \in [0, 1]$ is simply the normalized raw score of the benchmark for that specific datapoint.
>
> - for binary pass/fail benchmarks (e.g., GSM8K), $m_b(i)$ is 1 if correct, 0 if incorrect;
> - for benchmarks with continuous metrics (e.g., partial credit or scalar ratings), the score is normalized to the $[0, 1]$ range. Our metric $\chi_{model}^{(b,c)}$ computes the activation-weighted average of these scores.
>
> ---
>
> **> Q3. Benchmark gaps on competencies that are missing in the model itself?** This is an insightful question, and highlights a crucial distinction between activation and performance (correctness).
>
> Benchmark Gaps are calculated based on activations, not performance. Even if a model is "bad" at legal reasoning (i.e., it hallucinates or fails the logic), it will still likely activate legal concepts (e.g., "jurisdiction," "plaintiff") when processing the prompt. If the benchmark contains legal questions, the activations will be high, and we will not flag a benchmark gap, regardless of whether the model answers correctly. We believe that this is a strength of the method, and enables an analysis of benchmark gaps "through the eyes" of a model – it is a model's interpretation of the skills needed for a given benchmark. This is different from the competencies intended for evaluation by the benchmark designer.
>
> Model Gaps (Section 3.2) are where performance comes in. If the model activates the concept but gets the answer wrong, it will be flagged here.
>
> A theoretical failure mode is "concept blindness," where a model and subsequently its SAE both fail to activate a concept entirely because it was never learned, which would incorrectly appear as a benchmark gap. However, given that the SAEs are trained on massive, diverse corpora (16k+ concepts), we find that models almost always possess the internal representation (activation) even when they fail the task. Thus, "concept misuse" (high activation, low performance) is far more common than total "concept blindness" (zero activation).
>
> ---
>
> **> Q4. What are the base rate activations?** This is a great question. We control for the base rate effect specifically through the design of our Model Gap metric ($\chi_{model}^{(c)}$).
> As defined in Equation 4 (Section 3.2), the model performance score is calculated as follows:
>
> $$
> \chi_{model}^{(b,c)} = \frac{\sum m_b(i) \cdot \tilde{s}_{c,i}}{\sum \tilde{s}_{c,i}}
> $$
>
> By dividing the performance-weighted activations by the total activation mass of that specific concept ($\sum \tilde{s}_{c,i}$), the metric automatically normalizes for the concept's frequency. This ensures that our assessment of competency is decoupled from frequency.
>
> - **High Base-Rate Concept.** A concept that activates frequently (e.g., "common syntax") is averaged over thousands of instances.
> - **Low Base-Rate Concept.** A rare concept is averaged over its few active instances.
>
> If we were to "control" for base rates by artificially boosting rare concepts or penalizing frequent ones, we would distort the conditional probability of the model failing when that concept is relevant. Our method ensures that a 0% success rate on a rare safety concept is treated with the same severity as a 0% success rate on a common chat concept, which is critical for identifying safety risks that may be rare but catastrophic.

---

> > ### Comment · Reviewer_x8Ry · 2025-11-26
> >
> > Thank you to the authors for their thorough engagement with my comments and questions. I find the reliability and validity analyses quite compelling.
> >
> > It is a shame that the analysis with the Guard model was not achievable during this rebuttal period. Since it is only one model, it makes me question the usability of this method for its application on any model. Arguably, the method the authors propose should apply straightforwardly to any new model.
> >
> > Nonetheless, I think this is worthwhile methodology that deserves to see the light of day in the published literature. I therefore raise my score.

---

### Official Review · Reviewer_a3Lr · 2025-10-31

**Soundness:** 2
**Presentation:** 3
**Contribution:** 2
**Rating:** 2
**Confidence:** 4

**Summary:**

The paper introduces Competency Gaps (CG) — a novel evaluation framework leveraging Sparse Autoencoders (SAEs) to uncover two forms of evaluation shortcomings: Benchmark gaps (concepts underrepresented or missing in existing benchmarks) and Model gaps (concepts where models systematically underperform). By analyzing SAE concept activations across 10 benchmarks and 2 open-source LLMs (Gemma2-2B-Instruct and Llama3.1-8B-Instruct), the authors demonstrate that: Benchmarks overrepresent instruction-following and authority-related concepts. Models perform well on sycophantic or helpful concepts but poorly on boundary-maintaining, time reasoning, or rejection scenarios. The tool allows transparent exploration of benchmark and model concept coverage through an interactive web interface.

Key contributions:

- Methodological innovation: an SAE-based quantitative method for identifying benchmark and model coverage gaps.
- Empirical study: evaluation on 10 diverse benchmarks spanning reasoning, factuality, ethics, and math.

**Strengths:**

- This paper introduces a use of Sparse Autoencoders (SAEs) to analyze benchmark coverage and model competence at the concept level, moving beyond aggregate accuracy metrics. The framework provides a fresh interpretability-driven lens on how benchmark distributions shape perceived model performance—an idea with clear originality and conceptual significance.
- The use of sparse autoencoders to quantify benchmark and model “concept coverage” is well-formulated and technically sound, providing a coherent analytical framework.

**Weaknesses:**

- Experiments are restricted to two medium-sized instruction-tuned models (Gemma2-2B and Llama3.1-8B). Without results from smaller or larger models, the claimed “systematic competency gaps” may reflect architecture-specific or fine-tuning artifacts rather than generalizable phenomena.
- The proposed method assumes that SAE activations correspond to stable, human-interpretable concepts. However, the paper does not validate this assumption—for example, through multiple SAE runs, layer sensitivity, or concept consistency analysis. This dependency weakens the robustness of the theoretical foundation.
- While visualizations are compelling, there is no statistical significance testing or correlation analysis linking concept gaps to performance outcomes. The evidence remains descriptive rather than inferential, limiting confidence in the paper’s stronger claims.

**Questions:**

1. Can you demonstrate that modifying the composition of benchmark tests (e.g., removing overrepresented concepts or adding missing ones) actually changes model rankings or the measured capabilities?
2. The current experiments only include two medium-sized, instruction-tuned models. Have you tested smaller or larger models, as well as base models, to verify that the observed biases are generalizable?

---

> ### Author Response · Authors · 2025-11-22
>
> We thank the reviewer for their time and thoughtful review which helped us to strengthen our submission. We are glad to hear that the reviewer finds our approach “well-formulated and technically sound” and to give a  “fresh interpretability-driven lens on how benchmark distributions shape perceived model performance”. Moreover, we appreciate that the reviewer deems our contribution as demonstrating “clear originality and conceptual significance”.
>
> Below, we address the raised concerns point-by-point – and uploaded a revised manuscript where we tracked changes in red.
>
> ---
>
> **> Experiments Restricted to Two Medium-Sized Instruction-Tuned Models.** We greatly appreciate this point from the reviewer. It helped us realize that the original framing of our method may not have come across as intended. **We view our primary contribution as the method itself, which enables the unsupervised discovery of benchmark and model gaps.** The presented analysis on the two models and ten benchmarks serves as a first demonstration of the introduced method. Therefore, the made observations are not meant to reflect generalizable claims that hold across different architectures and different fine-tuning recipes, but solely reflect our findings on the two tested models. Applying the introduced methods across different models (i.e., different model architecture and different data) and analyzing whether there are generalizable competency gaps across models is indeed an interesting direction for future work enabled by the introduced method. In general, we envision that practitioners will use insights from our method iteratively to understand & refine benchmarks, as well as to understand & improve the strength and weaknesses of different models. → We have addressed this by refining the framing in the manuscript. Specifically, we have made changes to (a) the Abstract, (b) the list of contributions in Section 1: Introduction, and (c) the framing in Section 5: Results, and other locations throughout the paper.
>
> ---
>
> **> Assumption that SAE activations correspond to stable, human-interpretable concepts.** We thank the reviewer for pointing out this important aspect. The remark inspired us to run a  cross-SAE experiment verifying the robustness and generalizability of discovered SAE features. While the original paper evaluated Gemma2 2B and Llama 3.1 9B only using each model's own SAE respectively, we have now  run  a cross-SAE analysis where we analyze Gemma2 2B using Llama’s SAE, and vice versa. This experiment allows us to verify whether we would arrive at the same conclusions by using model-independent SAEs.  The insights remained consistent in these cross-SAE settings (despite the different SAE dictionary sizes).
>
> For example, comparing Llama on its own SAE yielded, as some of the worst-represented concepts, the following: (2874) Mathematical differentiation operators and notation and (13908) Numerical values, counts or measurements; meanwhile, analyzing the same model with an external SAE, specifically the one for Gemma 2, surfaced as some of the worst-represented concepts the following (2872) Explaining time requirements and duration and (9936) Dates and numeric sequences.
>
> This finding suggests that our method is able to yield meaningful insights even for models without a model-specific SAE, and further demonstrates the overall stability of the method. Nonetheless, we expect that a model-specific SAE will lead to the most precise and grounded results. These results have been added to the updated manuscript (Section 5.3, in particular Figure 3 and 4).
>
> ---
>
> **> Compelling Visualization But No Statistical Significance Testing or Correlation Analysis.** Motivated by this remark, we conducted two additional experiments to further probe the method’s robustness. First, we randomly removed 100 data points per benchmark and re-computed the CG metric across 100 repetitions, finding strong consistency and resilience to perturbation. For example, the standard deviations for the CG metrics while perturbing 20% of the benchmark data were 0.014 and 0.025 for CG_model and CG_bench, respectively. Second, we performed an adversarial perturbation designed to shift the overall CG score, and observed the expected movement. Both experiments are now described in Section 5.3.
>
> ---
>
> Please find the answers to the additional questions posed by the reviewer in the next comment.
>
> ---
>
> We are looking forward to further interactions with the reviewer and would appreciate to hear whether these clarifications and additional experimental evidence address their outstanding concerns. If so, and in consideration of the overall positive review by the reviewer, we would be happy if the reviewer would consider increasing the score.

---

> ### Author Response · Authors · 2025-11-22
>
> This response continues the discussion posted in our previous response to this review.
>
> ---
>
> **> Further Questions:**
>
> - **(Q1) Can you demonstrate that modifying the composition of benchmark tests (e.g., removing overrepresented concepts or adding missing ones) actually changes model rankings or the measured capabilities?** Thank you for suggesting this insightful experiment. We outline the detailed results in Section 5.3.
>
> - **(Q2) The current experiments only include two medium-sized, instruction-tuned models. Have you tested smaller or larger models, as well as base models, to verify that the observed biases are generalizable?** Please see our response under “Experiments Restricted to Two Medium-Sized Instruction-Tuned Models” and the additionally conducted robustness analysis. Taken all together, the proposed method can be applied to any model and benchmark, with consistent findings across different SAEs (according to the conducted robustness experiments).

---

> ### Comment · Reviewer_a3Lr · 2025-11-27
>
> Thank the authors for their responses. I have already increased my score.

---

### Official Review · Reviewer_wQzX · 2025-10-31

**Soundness:** 2
**Presentation:** 2
**Contribution:** 2
**Rating:** 4
**Confidence:** 4

**Summary:**

- This paper shows that current LLM benchmarks miss many concepts that models actually have, and thus fail to reveal their true weaknesses.
- Using SAEs, the authors map model behavior to fine-grained concepts to uncover gaps in both benchmarks and models.
- While benchmarks overtest abilities like instruction-following, models still struggle with polite refusal, boundary setting, and time reasoning, essentially the opposite of sycophancy, aligning well with the benchmark-gap findings.

**Strengths:**

**1. Introducing internal latent features for evaluation framework**

> This paper raises an interesting point that model evaluation does not always need to follow human-defined cognitive categories. Instead, it explores assessing capabilities through the model’s own latent features. Introducing SAEs in this context is a reasonable and promising direction, as it provides a way to diagnose models based on the representations they learn.

**2. Providing a meta-level perspective on benchmark coverage**

> It offers a useful way to see, from a broader and more meta perspective, how much of a model's overall capability space current benchmarks actually cover.

**Weaknesses:**

**1. Gap between stated motivation and actual research objective**

> Benchmarks are ultimately meant to communicate and compare core abilities at a broader level for a broad audience, not to enumerate every internal concept or serve primarily as a debugging tool. As a result, while this approach is useful for detailed model analysis, it does not resolve the aggregation problem highlighted in the motivation and is perceived as a diagnostic tool, rather than a practical framework for advancing benchmark design. **It is unclear how descriptive concept labels can inform high-level capability understanding or guide model improvement, particularly given the trade-offs across such highly granular features.**

**2. Lack of clarity on selecting latent features aligned with evaluation objectives**
> **The paper does not explain how to determine which latent features are meaningful and should be treated as evaluable capabilities.** Given that LLMs contain numerous internal dimensions that may not correspond to cognitive abilities, a principled criterion for selecting relevant features seems necessary. If such a mechanism is already discussed in the paper, clarification would be helpful.

**3. Missing discussion on SAE training data construction**

> Coverage of sparse concepts likely depends on constructing SAE training data that is as diverse as possible, ideally broad enough to approximate the full range of concepts encoded in the model itself, similar to the diversity of an LLM training corpus. However, the paper does not describe how the SAE training data was curated to ensure such conceptual diversity, which seems to be a crucial factor for achieving robust sparse-concept coverage.

**4. Need for insights beyond overall trend in benchmark coverage**

> The method shows the high-level trend of how much of a model's abilities current benchmarks cover (e.g., Figure 3, 4). This is useful, but for the tool to be more impactful, it would be helpful if it also suggested which kinds of abilities are missing and should be evaluated next to make benchmarks more balanced and fair. In other words, beyond showing the coverage trend, it would be valuable to connect these results to concrete capability areas that matter for evaluation.

**5. Model-dependent concept sets**

> Although each model can acquire different latent features, this approach does not provide a shared evaluation standard and therefore evaluates models relative to their own internal concept representations.

**Questions:**

- Conversely, are there capabilities that benchmarks such as BigGenBench (77 tasks) can evaluate but that do not emerge in the latent-based concept mapping? If so, could you share concrete examples?

- Could you elaborate on how such a large space of fine-grained conceptual units can be systematically organized and validated? Given that the absorption/common-pattern groupings are themselves highly granular, what principled structure do you envision for representing assessing these concepts in a coherent manner?

---

> ### Author Response · Authors · 2025-11-22
>
> We thank the reviewer for their time and thoughtful review which helped us to strengthen our submission. We are glad to hear that the reviewer found our approach "a useful way to see... how much of a model's overall capability space current benchmarks actually cover" and that using SAEs to diagnose models based on learned representations is a "promising direction."
>
> Below, we address the raised concerns point-by-point – and uploaded a revised manuscript where we tracked changes in red.
>
> ---
>
> **> Gap between stated motivation and actual research objective.** Thank you for the very thoughtful comment. Indeed, benchmarks are valuable in large part *because* they provide aggregation and summarization on a particular dimension(s), and we could have been clearer about our goals, which were not to overturn or displace this useful evaluation paradigm! Our goals are complementary to that paradigm: (a) to help identify model gaps that might not be obvious from such aggregated statistics (b) to help identify benchmark gaps, so that their aggregated scores can be more representative of the intended scope. To make this clearer in the manuscript itself, we have added clarification to the abstract (lines 28-31), and we have also added two boxes in Figure 2 with "workflow recommendations" on how to use the method (for identifying model and benchmark gaps). Please do let us know if you believe that further clarification is still needed.
>
> ---
>
> **> Lack of clarity on selecting latent features aligned with evaluation objectives.** Thank you for raising this question – we hope that the newly added "workflow recommendations" in Figure 2 will be helpful for adding clarity here as well. Part of the strength of the method is a reliance on SAE concepts which are many and wide-ranging, which allows the discovery of relevant gaps. At the same time, as you are hinting at, this multiplicity means that the method can return many results that may be difficult to sift through manually! Our recommendation is to use an LLM to cluster and filter through the gaps that are identified. This is the approach we used in our demonstrations of the method (Section 4). As validation of the approach, we found that the identified gaps did corroborate previously identified benchmark and model gaps, while also identifying novel gaps.
>
> ---
>
> **> Missing discussion on SAE training data construction.** We would like to clarify that we did not train the SAEs ourselves, but utilized high-quality pre-trained SAEs (specifically from the Gemma Scope and widely accepted open-source Llama SAEs) that were trained on massive, diverse corpora designed to approximate the full range of concepts encoded in the models. We will make this more prominent in the draft (Section 4.2.3 add pointer to specific section).
>
> ---
>
> **> Need for insights beyond overall trend in benchmark coverage.** We thank the reviewer for raising this point and suggesting alternative possibilities of impact and insights. Inspired by this comment, we:
> - Expanded the "Model Gaps" analysis. We revised Section 5: Results to focus much more heavily on specific concepts where models underperform, rather than just high-level coverage stats.
> - Added a new domain demonstration. We added an analysis of an arena-style benchmark (LMSYS Chatbot Arena) in the new Appendix D, demonstrating how the method identifies specific capability clusters in a different evaluation context.
> - Clarified the Workflow. The new Figure 2 explicitly maps out how a user moves from the "overall trend" to identifying specific missing abilities (e.g., polite refusal, boundary setting) to guide benchmark curation.
>
> ---
>
> **> Model-dependent concept sets.** Thank you for this important remark – we acknowledge that a transferability / robustness study was missing in the original manuscript. The remark inspired us to run a cross-SAE experiment verifying the robustness and generalizability of discovered SAE features. While we evaluated Gemma2 2B and Llama 3.1 9B using their model specific SAEs in the original draft, we extended the manuscript by a cross-SAE analysis where we analyze Gemma2 2B using Llama’s SAE, and vice versa. This experiment allows us to verify whether we would arrive at the same conclusions by using model-independent SAEs with their corresponding auto-interpretability labels. As described in the updated manuscript (section 5.3, in particular figure 3 and 4), the insights remained remarkably consistent across these cross-SAE settings despite the different SAE dictionary sizes. This finding suggests that our method is able to yield meaningful insights even for models without their model-specific SAE, and further demonstrates the overall stability of the method. Nonetheless, we expect that a model-specific SAE (if available) will lead to the most precise and grounded results.
>
> ---
>
> Please find the answers to the additional questions posed by the reviewer in the next comment.

---

> ### Author Response · Authors · 2025-11-22
>
> This response continues the discussion posted in our previous response to this review.
>
> ---
>
> **> Further Questions**
>
> - **(Q1) Conversely, are there capabilities that benchmarks such as BigGenBench (77 tasks) can evaluate but that do not emerge in the latent-based concept mapping? If so, could you share concrete examples?** Yes. While SAEs are excellent at decomposing inputs into granular semantic units, they might struggle to capture certain high-level, multi-step abstractions as a single "feature."
>
> - **(Q2) Could you elaborate on how such a large space of fine-grained conceptual units can be systematically organized and validated? Given that the absorption/common-pattern groupings are themselves highly granular, what principled structure do you envision for assessing these concepts in a coherent manner?** This is a very good question, and relates to the discussion above on "selecting latent features aligned with evaluation objectives". We have found that LLMs are quite effective at organizing and sifting through the results, in an objective-dependent way. We have included details on this methodology in Appendix Section E. We also recommend use of the web app (which we are open sourcing), which enables quick and easy inspection of the results. Future works could also explore the benefits of clustering and topic modelling on top of SAE features, as done in [1].
>
> [1] Zheng et. al., 2025, Model Directions, Not Words: Mechanistic Topic Models Using Sparse Autoencoders

---

### Official Review · Reviewer_Kt1z · 2025-10-31

**Soundness:** 3
**Presentation:** 1
**Contribution:** 2
**Rating:** 6
**Confidence:** 4

**Summary:**

The paper proposes to assess benchmarks and models in the conceptual space extracted by sparse autoencoders on its own representations. Having SAEs, the move is natural. This procedure surfaces familiar difficulties (palindromes, temporal reasoning) without supervision. Not a discovery, but a confirmation that the method is sound and that makes the results convincing.
The extreme skewedness of the concept distribution is an interesting insight.
As well as that the worst-performing concepts are often the opposites of these sycophantic behaviors, that was expected, but given the unsupervised nature of the method, it is remarkable.

**Strengths:**

It is a tool that could turn useful to people in the interpretability field.
As stated before, it confirms expected behaviors without any supervision, supporting the soundness of the method...

...but at the same time in this paper does not emerge anything particularly new.

The latter is not a Strength, but I think also not a Weakness. Honestly I find this structured format of reviews annoying, it is more difficult to write an organic judgement.

**Weaknesses:**

The main problem I have with this work is its writing. It is heavy. The overabundance of references sometimes interrupts the conceptual through-line. Papers should be written to be read, not merely to accompany code, and here the balance leans too far towards the latter.

Also the work remains confined to its enclave. It does not try to speak beyond those already initiated into SAEs, and the writing reflects this inward posture, should be more self-contained, and just more pleasant to read. A paper should be an invitation to thought. Here, I don't feel invited.

For reference, this is a well written paper in the interpretability area: https://transformer-circuits.pub/2025/attention-qk/index.html

**Questions:**

Now I guess I should write my conclusions in the Questions section, since there is no other section afterwards.

The contribution is real, if modest. A small community will find this tool useful, its functioning is sound and well justified. It is not well presented but indeed its ambition was not to reach a greater public.

The unfortunate fact is that I am part of the greater public, and I also have the opinion that any paper should be an invitation to thought for a large enough community.

Nevertheless, my recommendation is a poster. I would not feel good in seeing this paper rejected. I would put 7. There is no 7 so I put 6 and will change to 8 later. I will do it anyways, but would appreciate if the authors make the paper more accessible and pleasant to read, for example with a brief explanation of SAEs.

---

> ### Author Response · Authors · 2025-11-22
>
> We thank the reviewer for their honest feedback, including the rarely voiced sentiment that a paper should be an "invitation to thought". We have made significant changes to the framing and writing of the paper, and we hope that this improves the delivery.
>
> In terms of the results and methodology, we are glad the reviewer found the method to be "sound" and the results to demonstrate an "interesting insight". Below, we describe how we have overhauled the manuscript to address the concerns regarding writing style and accessibility.
>
> ---
>
> **> Writing and number of references.** We have significantly revised the text to improve flow and readability. We stripped away unnecessary jargon and reduced the density of interruption-heavy citations in the narrative sections. We have reframed the paper to assume little to no prior exposure to the niche mechanics of SAEs, explaining the basics in a more accessible way, and focusing more on the practical implications of the findings.
>
> ---
>
> **> Making the paper accessible to those unfamiliar with SAEs.** To make the paper self-contained for the broader community, we have:
>
> - Added an SAE Primer. We added a dedicated explanation of Sparse Autoencoders starting in Section 1, line 52. This section intuitively explains what SAEs are and why they are useful for evaluation, without bogging the reader down in excessive math too early.
> - Broadened the Scope. To ensure the paper speaks to the a broader audience, we added Appendix D, which applies our method to the LMSYS Chatbot Arena, which is familiar to most folks in the field, and has been driving much of the progress and discussion around model evals.
> - Added Practical Workflow Recommendations. Figure 2 now illustrates how CG can be used in a very practical fashion to iterate on the quality of a model or a benchmark (explained further below).
>
> ---
>
> **> Making the contribution relevant for the wider community.** We want to clarify the ambition of the paper: we do want to reach the greater public. To operationalize this, we added a new Figure 2 with recommended workflows.
>
> This figure is designed for the general practitioner. It moves away from abstract theory and provides a step-by-step guide on how to:
> - Evaluate a Model. Using our metrics to find gaps and then using an LLM to cluster those gaps into human-readable themes.
> - Build a Benchmark. Using the tool to identify what data is missing from an evaluation suite.
>
> We hope this visual guide transforms the paper from a "companion to code" into a more practical methodology for better evals.
>
> ---
>
> **> Overall recommendation.** We are committed to earning that score change, if deserved. We have worked hard to make the paper more "pleasant to read"  by streamlining the narrative and adding the background necessary for a general audience.
>
> We are very open to further collaboration on this front. If, after looking at the revised manuscript, you feel there are still specific sections that feel "heavy" or "confined," please let us know. We are eager to make the necessary adjustments to ensure this work caters to the broader community.

---

### Author Response · Authors · 2025-11-22
**Thank you for your thoughtful feedback!**

We thank the reviewers for their thoughtful feedback, and are encouraged by the positive commendations of our results as "remarkable", "useful", "promising", "“well-formulated and technically sound”, and a "novel and innovative contribution".

We also thank the reviewers for incredibly helpful suggestions. Thanks to your suggestions, we have now made significant additions and updates to the manuscript (highlighted in red), including running a significant number of new experiments. These include:

- Reframing our contributions to highlight and clarify the main contributions, which are the method, the kinds of insights it can provide, and how they can be practically operationalized. Simultaneously, we have emphasized that the specific results for Gemma 2 and Llama 3.1 serve primarily as demonstration case studies for the method.
- Expanding the robustness analysis of our method via two new sets of experiments. Firstly, we have verified that – even when the SAEs used for the analysis are swapped (e.g., the Gemma SAE for analyzing Llama) – the insights are remarkably consistent. We have also performed random and adversarial perturbations.

Below are more detailed responses for each review. We believe that the paper is now significantly stronger; we appreciate your time and thoughtfulness in helping us achieve that.

---

### Meta-Review · Area_Chair_TTCZ · 2026-01-03

**Summary:**

While all reviewers agree that the proposed framework would be impactful for the community to better understand evaluation, it has the following two major concerns and a few minor ones.

1. The proposed method and the concern are not new (Kt1z), and the proposed approach (which relies on sparse autoencoders; SAEs) does not fundamentally address the problem (wQzX, a3Lr, x8Ry). I share the reviewers' concerns about the reliability and generalizability of the presented SAE results, and I find the evaluation of the proposed framework insufficient---for example, the coverage and the reliability of the concepts are both questionable (wQzX, a3Lr).
2. The presentation could be improved a lot (Kt1z, wQzX).
3. Some minor issues, such as missing references and robustness checking, have been addressed in the rebuttal.

**Reviewer Concerns:**

I would like to note that this paper has some merit, such as re-emphasizing an important problem and proposing a promising framework to address it. While the minor concerns have been addressed, and the authors' rebuttal has also addressed some of the coverage and reliability concerns, I don't think the paper, after rebuttal, is convincing enough to justify the claim that SAEs could meaningfully contribute to understanding the competence gap.

**Reviewer Scores:**

Reviewers a3Lr and x8Ry have mentioned that they have improved their scores (although I don't have the final scores). I think Kt1z is unlikely to improve their score, since their main concern is the confusion caused by the presentation, which I partially agree with. I don't have an educated guess about Reviewer wQzX---maybe their score will rise from a borderline reject to borderline accept.

---

### Decision · Program_Chairs · 2026-01-26

Reject